# Step-Size Stability in Stochastic Optimization: A Theoretical Perspective

**Fabian Schaipp** [1]   **Robert M. Gower** [2]   **Adrien Taylor** [1]

## Abstract

We present a theoretical analysis of stochastic optimization methods in terms of their sensitivity with respect to the step size. We identify a key quantity that, for each method, describes how the performance degrades as the step size becomes too large. For convex problems, we show that this quantity directly impacts the suboptimality bound of the method. Most importantly, our analysis provides direct *theoretical* evidence that adaptive step-size methods, such as SPS or NGN, are more robust than SGD. This allows us to quantify the advantage of these adaptive methods beyond empirical evaluation. Finally, we show through experiments that our theoretical bound qualitatively mirrors the actual performance as a function of the step size, even for non-convex problems.

## 1. Introduction

Training machine learning models requires algorithms that are at the same time efficient and robust. Several studies have shed light on the effect of subtle choices in architecture, loss function or optimizer on training efficiency and stability (Wortsman et al., 2024; Everett et al., 2024; OLMo et al., 2025). Specifically Wortsman et al. (2024) focus on the effect on the range of learning rates, for which training proceeds smoothly. This is not surprising, as it is widely accepted in the optimization and machine learning community that the learning rate is arguably the most important hyperparameter to tune (Bengio, 2012).

For this paper, we will focus on the effect of the *optimization algorithm* on training stability. In the past, several methods have been designed to facilitate or even fully avoid the issue of tuning the learning rate (more in Section 2 below). While empirical evaluation in machine learning is closely

paying attention to the issue of learning-rate sensitivity, optimization theory on the other hand puts more focus on convergence rates (where it is sufficient that the optimal rate is attained for *some* step size).

To fill in this gap, here we compare – with a theoretical framework – how robust several *stochastic* optimization methods are with respect to their step size. In particular, we focus on stochastic gradient descent (SGD), the stochastic Polyak step size (SPS), non-negative Gauss-Newton steps (NGN), and stochastic proximal point (SPP). Throughout the paper we will use the term *sensitivity* to refer to the final loss/suboptimality as a function of (the scale of) the step size: for small step sizes, all methods will only progress slowly; for large step sizes however, some methods behave more stably than others.

We summarize our contributions below.

**Summary and contributions.**

- We identify a key quantity within our theoretical framework, which we call the *stability index*, for determining how the method's suboptimality behaves as a function of the step size. Our theory covers a general family of model-based methods and applies to SGD, SPS SPP, and partially NGN in the convex and weakly convex setting.

- Previous *empirical* results suggest that SPS, NGN, and SPP are always less sensitive to large step sizes compared to SGD. We will show that this is provably true. In particular, our results show that NGN **and** SPS **are always at least as stable as** SGD. For SPS, this holds for any guess of the optimal value/lower bound; interpolation is not necessary to obtain an advantage, but it helps. Our analysis also reveals the key quantities to determine the gain in stability, which are related to interpolation and the under-estimation of the optimal (batch) loss.

- In several convex and non-convex toy experiments (linear/logistic regression, training a ResNet on CIFAR10), we track the theoretical quantities previously obtained to explain the improved stability of SPS, NGN, and SPP. We show that the actual performance and the theoretical suboptimality bound match closely

[1]Inria, Departement d'Informatique de l'Ecole Normale Superieure, PSL Research University, Paris, France [2]CCM, Flatiron Institute, New York City. Correspondence to: Fabian Schaipp <fabian.schaipp@inria.fr>.

*Proceedings of the 43rd International Conference on Machine Learning*, Seoul, South Korea. PMLR 306, 2026. Copyright 2026 by the author(s).

in predicting the range of learning rates that give good performance. Surprisingly, this is also true for non-convex problems, where our theory is not applicable.

**Limitations.** The goal of this paper is predominantly to explain empirically observed phenomena with a general theoretical framework. Even though our experiments suggest a close match between theory and practice, we leave it open how to exploit this for practical applications. Further, some parts of our theory are restricted to convex problems, and only to a specific family of methods; in particular, it does not cover momentum, or preconditioning methods such as `Adam`, which are prevalent in practice.

**Setup and background.** We consider the training problem

$$\min_{x \in \mathbb{R}^d} f(x), \quad f(x) := \mathbb{E}_s[f(x, s)]. \tag{1}$$

Here, $x \in \mathbb{R}^d$ are the learnable parameters of a machine learning model, and $f$ is the loss function. In $f$, the expectation is taken over the distribution of a random variable $s$ that maps to the space or set $\mathcal{S}$ (typically the training set). We assume that $f(\cdot, s)$ has a suitable subdifferential for every $s \in \mathcal{S}$ (for example, see Rockafellar (1970); Clarke (1983); Bolte & Pauwels (2021)). We denote elements of the subdifferential by $g \in \partial f(x, s)$. We assume that problem (1) admits a solution $x_\star$ and denote $f_\star := f(x_\star)$.

We briefly introduce the four methods we will mainly focus on throughout: for a step size $\alpha > 0$[1], *stochastic gradient descent* is given by

$$x_{t+1} = x_t - \alpha g_t, \quad g_t \in \partial f(x_t, s_t). \tag{SGD}$$

It usually becomes unstable when the magnitude of $\alpha > 0$ increases. It has been shown empirically that other methods are much less sensitive to large values of $\alpha$: in particular, this is known for the *stochastic Polyak step size* (Polyak, 1987; Asi & Duchi, 2019a; Loizou et al., 2021): for a known lower bound $C_t \leq \inf_{z \in \mathbb{R}^d} f(z, s_t)$, let $g_t \in \partial f(x_t, s_t)$, then

$$x_{t+1} = x_t - \min\{\alpha, \frac{f(x_t, s_t) - C_t}{\|g_t\|^2}\} g_t. \tag{SPS}$$

The above method allows for much larger $\alpha$ than `SGD` across many problem instances in machine learning (Schaipp et al., 2023; 2024). Closely related to the Polyak step size is the non-negative Gauss-Newton step (Orvieto & Xiao, 2024). In the stochastic setting, with $g_t \in \partial f(x_t, s_t)$, it is given by

$$x_{t+1} = x_t - \frac{\alpha}{1 + \frac{\alpha}{2f(x_t, s_t)} \|g_t\|^2} g_t. \tag{NGN}$$

Finally, it has also been shown empirically that the stochastic proximal point method, given by

$$x_{t+1} = \arg\min_{y \in \mathbb{R}^d} f(y, s_t) + \frac{1}{2\alpha} \|y - x_t\|^2, \tag{SPP}$$

is less sensitive to the choice of $\alpha$ than `SGD` (Asi & Duchi, 2019a; Davis & Drusvyatskiy, 2019; Milzarek et al., 2024).

In convex, nonsmooth optimization, we can usually derive bounds[2] of the form

$$\mathbb{E}[f(x_T) - f_\star] \leq \frac{\mathcal{T}_1}{\alpha} + \mathcal{T}_2(\alpha),$$

where the term $\mathcal{T}_1/\alpha$ can be seen as a bias term; usually we have that $\mathcal{T}_1 \propto 1/T$. However, the sensitivity to large step sizes is determined by the second term $\mathcal{T}_2(\alpha)$, reflecting the variance; for `SGD` this term grows linearly with $\alpha$.

We will show that for other methods such as `SPS`, `NGN`, and `SPP` the scaling of $\mathcal{T}_2(\alpha)$ in $\alpha$ is much more benign. Further, we will show through experiments that the theoretical bound closely reflects the actual performance with respect to learning-rate tuning. This gives a theoretical explanation for the empirically observed improvements in stability for `SPS`, `NGN`, and `SPP`.

**Notation.** Unless explicitly stated otherwise, $\|\cdot\|$ and $\langle \cdot, \cdot \rangle$ denote the standard Euclidean norm and its scalar product. Throughout we will refer to $\alpha$ (and later $\alpha_t$) as the *step size*, in the sense that this parameter has to be user-specified/tuned in practice. However, note that for the update rules of (SPS) and (NGN) the *effective* step size, i.e., the multiplier of $g_t$, is different from $\alpha$ in general.

## 2. Related Work

**Model-based stochastic optimization.** The core concept of model-based optimization is, in each iteration, to construct a simplified model/surrogate of the objective, and to minimize this surrogate in a proximal fashion. Theory for this family of methods has been established in the seminal works of Asi & Duchi (2019a) and Davis & Drusvyatskiy (2019). Concretely, Asi & Duchi (2019a;b) prove that models which share a certain lower-bound condition[3] lead to methods that are stable, in the sense that the iterates are almost surely bounded for square-summable step sizes (Asi & Duchi, 2019a, Cor. 3.5). However, this result does not quantify how performance degrades as the scale of the step sizes grows. Moreover, the convergence result that follows from stability (Asi & Duchi, 2019a, Prop. 3.8) is asymptotic. We refine this analysis by looking at non-asymptotic bounds, and their scaling with respect to the step size $\alpha$.

---

[1]For now we assume a constant step size purely to keep the presentation simple; all results later work for time-dependent step sizes $\alpha_t$.

[2]For the concrete realization of such a bound, see Theorem 3.4.
[3]For details, see condition (C.iv) in Asi & Duchi (2019a).

**The issue of learning-rate tuning.** Stochastic versions of the Polyak step size come in many different flavors (Berrada et al., 2019; Loizou et al., 2021; Gower et al., 2022). However, most of the practical implementations cap the effective step size with a hyperparameter $\alpha$ as depicted in equation (SPS). This hyperparameter can be interpreted as a user-specified learning rate. Versions of SPS, combinable with weight decay, momentum, and preconditioning, have shown promising empirical performance across many deep learning tasks, often being much less sensitive to the choice of $\alpha$ (Schaipp et al., 2023; 2024). SPS usually performs well across multiple orders of magnitude for $\alpha$. In this paper, we provide a theoretical analysis of this remarkable stability property of SPS.[4]

Similar to the Polyak step size, the non-negativity of loss functions in machine learning has been leveraged to derive Gauss-Newton type step sizes (Orvieto & Xiao, 2024; Islamov et al., 2025). The resulting method NGN has been proven to be similarly robust for deep learning tasks as Polyak-based methods (Islamov et al., 2025).

In this paper, we will focus mainly on SPS and NGN. However, we want to briefly summarize other efforts to design optimization algorithms that need less (or no) tuning of the step size (these are sometimes subsumed under the name "parameter-free algorithms"). One line of work sets the learning rate on-the-fly by estimating a-priori unknown problem parameters based on the optimization trajectory (Carmon & Hinder, 2022; Defazio & Mishchenko, 2023; Ivgi et al., 2023; Mishchenko & Defazio, 2024). Other parameter-free methods based on coin betting entirely remove the learning rate as hyperparameter (Orabona & Pál, 2016; Orabona & Tommasi, 2017). Recently, Kasimbeg et al. (2025) have conducted an empirical comparison of several of the methods mentioned above.

**Stochastic proximal point.** For deterministic problems, it is well known that the proximal point method (Martinet, 1970; Rockafellar, 1976) converges for any sequence of step size, and potentially arbitrarily fast.[5] In the stochastic case, SPP has been analyzed in various settings (Bertsekas, 2011; Boyd & Ryu, 2014; Patrascu & Necoara, 2017; Asi & Duchi, 2019a; Davis & Drusvyatskiy, 2019; Milzarek et al., 2024). However, to the best of our knowledge, no clear theoretical advantage of SPP has been established in the stochastic setting. Here, we prove an improvement in step-size stability for SPP, and show how it is closely related to interpolation properties.

---

[4]In Section B.2, we comment on why previous analyses such as the one by Loizou et al. (2021) do not fully reflect the empirical results.

[5]The caveat: a faster rate comes with a larger step size, which makes the subproblems in general harder to solve.

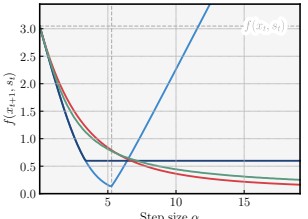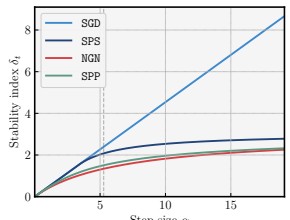

*Figure 1.* Illustration of theory for $f(x, s_t) = \ln(1 + \exp(-x)) + \max\{x - 2, 0\}$ and $x_t = -3$. **(Left)** Next-iterate loss as a function of step size $\alpha$. **(Right)** Stability index $\delta_t$ as a function of $\alpha$. Vertical line marks best SGD step size. For large $\alpha$, stable loss values coincide with benign ($\approx$ sub-linear) increase of $\delta_t$.

## 3. Theoretical Analysis

We study model-based stochastic proximal point with step sizes $(\alpha_t)_{t \in \mathbb{N}} \subset \mathbb{R}_{>0}$. Let $y \mapsto f_x(y, s)$ be a model of $f(\cdot, s)$ around $x \in \mathbb{R}^d$, then the update is given by

$$x_{t+1} = \arg\min_{y \in \mathbb{R}^d} f_{x_t}(y, s_t) + \frac{1}{2\alpha_t} \|y - x_t\|^2. \quad (2)$$

We define $\delta_t$, called from now on the **stability index**, as

$$\boxed{\delta_t := f(x_t, s_t) - f_{x_t}(x_{t+1}, s_t) - \frac{1}{2\alpha_t} \|x_{t+1} - x_t\|^2.} \quad (3)$$

We will later show that $\delta_t$, more precisely its scaling with respect to $\alpha_t$, determines how stable the model-based update (2) is to the choice of the step size $\alpha_t$. Fig. 1 illustrates this phenomenon on a toy example. Before we derive $\delta_t$ for concrete choices of $f_x(\cdot, s)$ (see Section 4), we start by presenting the role of $\delta_t$ in suboptimality (resp. stationarity) bounds for the convex (resp. weakly convex) case.

### 3.1. Convex Case

In this sub-section, we make the following model assumptions: for every $s \in \mathcal{S}$, it holds:

(A1) The mapping $y \mapsto f_x(y, s)$ is convex.

(A2) We have $f_x(x, s) = f(x, s)$ for all $x \in \mathbb{R}^d$ and $f_x(y, s) \le f(y, s)$ for all $y \in \mathbb{R}^d$.

We will show later that if $f(\cdot, s)$ is convex, for many methods considered here, the above two assumptions are satisfied (see Section 4). Further, from the definitions (3) and (2), it follows immediately that the stability index is always non-negative under (A2).

**Lemma 3.1.** *If the model satisfies $f_x(x, s) = f(x, s)$, in particular if (A2) holds, then $\delta_t \ge 0$ holds for any $x_t \in \mathbb{R}^d, s_t \in \mathcal{S}$ and any $\alpha_t > 0$.*

Note that the model-based framework implicitly anchors the step size of all methods onto the same scale, therefore

making them comparable: this is due to update (2), and the first part of (A2).

Next, we state the base lemma for our theoretical analysis.

**Lemma 3.2.** *Let $1 \leq k \leq T$ and let $u \in \mathbb{R}^d$ be measurable with respect to $x_k$. Let the iterates $(x_t)_{t \in \mathbb{N}}$ be generated by (2). Assume that (A1)-(A2) hold. Denoting $D_t := \|x_t - u\|^2$, we have*

$$\sum_{t=k}^{T} \alpha_t \mathbb{E}[f(x_t) - f(u)] \leq \frac{1}{2}\mathbb{E}[D_k - D_{T+1}] + \sum_{t=k}^{T} \alpha_t \mathbb{E}[\delta_t].$$

*Proof.* Under (A1), Asi & Duchi (2019a, Lem. 3.7) show

$$\frac{1}{2}\|x_{t+1} - u\|^2 \leq \frac{1}{2}\|x_t - u\|^2 - \frac{1}{2}\|x_{t+1} - x_t\|^2$$
$$-\alpha_t[f_{x_t}(x_{t+1}, s_t) - f_{x_t}(u, s_t)].$$

Now, we decompose

$$f_{x_t}(x_{t+1}, s_t) - f_{x_t}(u, s_t) =$$
$$\underbrace{f(x_t, s_t) - f_{x_t}(u, s_t)}_{=: \, U_1} + \underbrace{f_{x_t}(x_{t+1}, s_t) - f(x_t, s_t)}_{=: \, U_2}.$$

Due to (A2) we have $U_1 \geq f(x_t, s_t) - f(u, s_t)$. For the other term, use (3) to obtain

$$-\alpha_t U_2 - \frac{1}{2}\|x_{t+1} - x_t\|^2 = \alpha_t \delta_t.$$

Altogether, we get

$$\frac{1}{2}(D_{t+1} - D_t) \leq -\alpha_t[f(x_t, s_t) - f(u, s_t)] + \alpha_t \delta_t. \quad (4)$$

Then apply conditional expectation, and sum over $t = k, \ldots, T$. $\square$

**Theorem 3.3** (Average-iterate bound). *Assume that (A1)-(A2) hold. Let $T \in \mathbb{N}$ and let the iterates $(x_t)_{t \in \mathbb{N}}$ be generated by (2). Let $x_\star \in \mathbb{R}^d$ and $D := \|x_1 - x_\star\|$. Then, it holds, for both $\bar{f}_T = \mathbb{E}[f((\sum_{t=1}^{T} \alpha_t)^{-1} \sum_{t=1}^{T} \alpha_t x_t)]$ and $\bar{f}_T = \min_{t=1,\ldots,T} \mathbb{E}[f(x_t)]$ that*

$$\bar{f}_T - f(x_\star) \leq \frac{D^2}{2\sum_{t=1}^{T} \alpha_t} + \frac{\sum_{t=1}^{T} \alpha_t \mathbb{E}[\delta_t]}{\sum_{t=1}^{T} \alpha_t}.$$

*Proof.* Apply Lemma 3.2 with $k \to 1$ and $u \to x_\star$. Then, divide by $\sum_{t=1}^{T} \alpha_t$ (and apply Jensen's inequality). $\square$

**Theorem 3.4** (Last-iterate bound). *Assume that (A1)-(A2) hold. Let $T \in \mathbb{N}$ and let the iterates $(x_t)_{t \in \mathbb{N}}$ be generated by (2). Let $x_\star \in \mathbb{R}^d$ and $D := \|x_1 - x_\star\|$. Then, it holds*

$$\mathbb{E}[f(x_T) - f(x_\star)] \leq \frac{D^2}{2\sum_{t=1}^{T} \alpha_t} + \frac{\sum_{t=1}^{T} \alpha_t \mathbb{E}[\delta_t]}{\sum_{t=1}^{T} \alpha_t}$$
$$+ \sum_{k=1}^{T-1} \frac{\alpha_k}{\sum_{t=k+1}^{T} \alpha_t}\left(\frac{1}{\sum_{t=k}^{T} \alpha_t}\sum_{t=k}^{T} \alpha_t \mathbb{E}[\delta_t]\right).$$

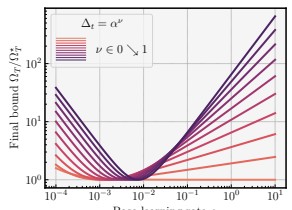 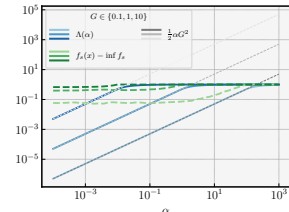

*(a)* Illustration of Theorem 3.4     *(b)* PEPit verification

*Figure 2.* **(Left)** Stability with respect to learning-rate $\alpha$ under different scalings of $\Delta_t$. **(Right)** Values of $\Lambda(\alpha)$ **(blue)** computed with PEPit, for different Lipschitz constants $G$. We plot the upper bound $f_s(x) - \inf f_s$ for the worst-case instance in **(green)**, and the corresponding $\delta^{\text{SGD}} = \frac{1}{2}\alpha G^2$ **(grey)**.

*Proof.* Follows identically to the proof of Defazio et al. (2023, Thm. 10) (or Schaipp et al. (2025, Thm. 3.1)), using Lemma 3.2 instead of Schaipp et al. (2025, Lem. E.1), and replacing $\frac{1}{2}\eta_t^2 \mathbb{E}\|g_t\|^2$ by $\alpha_t \mathbb{E}[\delta_t]$. $\square$

Throughout, we will denote the right hand side of the bound in Theorems 3.3 and 3.4 as follows:

$$\bar{f}_T - f(x_\star) \leq \Omega_T^{\text{avg}}, \quad \mathbb{E}[f(x_T) - f(x_\star)] \leq \Omega_T^{\text{last}}.$$

**Illustration of Theorem 3.4.** Before deriving the value of $\delta_t$ for the methods of interest, we illustrate how the bound $\Omega_T^{\text{last}}$ in Theorem 3.4 (analogously Theorem 3.3) reflects the aspect of stability with respect to large step sizes $\alpha_t$ (see Fig. 2a). For this, we fix a constant step size $\alpha_t = \alpha$.

As discussed above, the scaling behavior in $\alpha$ depends crucially on the scaling of $\mathbb{E}[\delta_t]$ in $\alpha$. Purely for the sake of illustration, we now make the simplification $\Delta_t := \mathbb{E}[\delta_t] = \alpha^\nu$ for some $\nu \geq 0$. Denote $H_T := \sum_{s=1}^{T} \frac{1}{s}$. Then, with $\alpha_t = \alpha$ and $\Delta_t = \alpha^\nu$, we have

$$\Omega_T^{\text{last}} = \frac{D^2}{2\alpha T} + \alpha^\nu[1 + H_{T-1}].$$

> **Takeaway:** The stability of the bound $\Omega_T$ with respect to large learning rates $\alpha$ crucially depends on the scaling of $\Delta_t = \mathbb{E}[\delta_t]$ (as illustrated with our simplification $\Delta_t = \alpha^\nu$, see Fig. 2a).

### 3.2. Weakly Convex Case

This section shows that the role of $\mathbb{E}[\delta_t]$ is not limited to the convex case. For this, we first define a function $\varphi : \mathbb{R}^d \to \mathbb{R}$ to be $\rho$-weakly convex for $\rho \geq 0$ if $\varphi + \frac{\rho}{2}\|\cdot\|^2$ is convex. In this case, for any subgradient $u \in \partial\varphi(x)$ it holds $\varphi(y) \geq \varphi(x) + \langle u, y - x \rangle - \frac{\rho}{2}\|y - x\|^2$ for all $y \in \mathbb{R}^d$ (Davis & Drusvyatskiy, 2019, Lem. 2.1). If $\varphi$ is $\rho$-weakly convex, its

*Moreau envelope* (Moreau, 1965) is given by

$$\mathrm{env}_\varphi^\alpha(x) := \min_{y \in \mathbb{R}^d} \varphi(y) + \frac{1}{2\alpha}\|y - x\|^2, \quad \alpha < 1/\rho.$$

The Moreau envelope is differentiable, and $\|\nabla\mathrm{env}_\varphi^\alpha(x)\|$ can be interpreted as a measure of near-stationarity of $\varphi$ at $x$ (see Davis & Drusvyatskiy (2019, Section 2.2)).

We replace (A1) and (A2) with their natural generalization:

(Ã1) For any $x \in \mathbb{R}^d$ and $s \in \mathcal{S}$, the mapping $y \mapsto f_x(y, s)$ is $\lambda$-weakly convex for some $\lambda \geq 0$.

(Ã2) We have $f_x(x, s) = f(x, s)$ for all $x \in \mathbb{R}^d$. Further, for all $s \in \mathcal{S}$ there exists $\rho_s \geq 0$ such that for all $x, y \in \mathbb{R}^d$ we have $f_x(y, s) \leq f(y, s) + \frac{\rho_s}{2}\|x - y\|^2$. Assume that $\rho := \mathbb{E}_s[\rho_s] < +\infty$.

Note that if (Ã1) and (Ã2) are satisfied with $\lambda = \rho_s = 0$, then (A1) and (A2) are satisfied. Further, (Ã1)-(Ã2) imply that $f = \mathbb{E}[f(\cdot, s)]$ itself is $(\rho + \lambda)$-weakly convex (Davis & Drusvyatskiy, 2019, Lem. 4.1).

Below, we state analogous results to Lemma 3.2 and Theorem 3.3 for the weakly convex case. Proofs are deferred to Section A.1 in the Appendix.

**Lemma 3.5.** *Assume that (Ã1) and (Ã2) hold. Let $\alpha_t \in (0, 1/\lambda)$. Let $x_t \in \mathbb{R}^d$ and denote by $\mathbb{E}_t$ the conditional expectation with respect to the filtration generated by $\{x_1, \dots, x_t\}$. Let $\bar{\rho} > \rho + \lambda$ and let $\hat{x}_t = \mathrm{prox}_{(1/\bar{\rho})f}(x_t)$. Then, we have*

$$\mathbb{E}_t\|x_{t+1} - \hat{x}_t\|^2 \leq \|x_t - \hat{x}_t\|^2$$
$$- \frac{\alpha_t(\bar{\rho} - \rho - \lambda)}{1 - \alpha_t\lambda}\|x_t - \hat{x}_t\|^2 + \frac{2\alpha_t}{1 - \alpha_t\lambda}\mathbb{E}_t[\delta_t]. \quad (5)$$

**Theorem 3.6.** *Assume that (Ã1) and (Ã2) hold, which in particular implies that $f$ is $(\rho + \lambda)$-weakly convex. Let $1 \leq k \leq T$ and let the iterates $(x_t)_{t \in \mathbb{N}}$ be generated by (2). Let $\alpha_t \in (0, 1/\bar{\rho})$ for all $t \in \mathbb{N}$ for some $\bar{\rho} > \rho + \lambda$. Denoting $\mathcal{E}_t := \mathbb{E}[\mathrm{env}_f^{1/\bar{\rho}}(x_t)]$, it holds*

$$\sum_{t=k}^{T} \frac{\alpha_t}{1 - \alpha_t\lambda}\mathbb{E}\|\nabla\mathrm{env}_f^{1/\bar{\rho}}(x_t)\|^2 \leq \frac{2\bar{\rho}}{\bar{\rho} - \rho - \lambda}\left[\mathcal{E}_k - \mathcal{E}_{T+1}\right]$$

$$+ \frac{2\bar{\rho}^2}{\bar{\rho} - \rho - \lambda}\sum_{t=k}^{T}\left(\frac{\alpha_t}{1 - \alpha_t\lambda}\mathbb{E}[\delta_t]\right).$$

The above bound can be easily turned into a convergence rate for the near-stationarity measure $\mathbb{E}\|\nabla\mathrm{env}_f^{1/\bar{\rho}}(\cdot)\|^2$, for example see (Davis & Drusvyatskiy, 2019, Thm. 4.3). We can see that the role of $\mathbb{E}[\delta_t]$ in Theorem 3.6 is analogous to the previous results in Section 3.1.

## 4. Model Choices

In this section, we derive the stability index (3) for classical model choices. Proofs are deferred to Section A.2 in the Appendix. Let the *regularized model* be given by

$$\Psi_t := f_{x_t}(x_{t+1}, s_t) + \frac{1}{2\alpha_t}\|x_{t+1} - x_t\|^2, \quad (6)$$

hence $\Psi_t = f(x_t, s_t) - \delta_t$. We assume from now on that $f(\cdot, s)$ is $\eta_s$-weakly convex, that is, there exists $\eta_s \geq 0$ such that for all $x, y \in \mathbb{R}^d$ it holds

$$f(y, s) - f(x, s) \geq \langle g, y - x \rangle - \frac{\eta_s}{2}\|y - x\|^2 \quad (7)$$

for all $g \in \partial f(x, s)$ and all $s \in \mathcal{S}$.

### 4.1. Linear Model (SGD)

If $f_x(y, s) = f(x, s) + \langle g, y - x \rangle$ for some $g \in \partial f(x, s)$, then $x_{t+1} = x_t - \alpha_t g_t$ and using this model in (6) gives $\Psi_t = f(x_t, s_t) - \frac{\alpha_t}{2}\|g_t\|^2$. Therefore, we have $\delta_t^{\mathrm{SGD}} = \frac{\alpha_t}{2}\|g_t\|^2$ which recovers Schaipp et al. (2025, Thm. 3.1). It is easy to see that (Ã1) holds with $\lambda = 0$ and (Ã2) holds with $\rho_s = \eta_s$ due to (7).

### 4.2. Truncated Model (SPS)

Given access to a lower bound on our stochastic function $C_s \leq \inf_z f(z, s)$ we can define a *truncated model*

$$f_x(y, s) = \max\{f(x, s) + \langle g, y - x \rangle, C_s\} \quad (8)$$

for some $g \in \partial f(x, s)$. This model satisfies (Ã1) with $\lambda = 0$ and (Ã2) with $\rho_s = \eta_s$ (Schaipp et al., 2023, Lem. 2). This truncated model also gives a stability index that is always smaller or equal than the stability index of SGD.

**Lemma 4.1.** *Let $s \mapsto C_s$ be a mapping such that $C_s \leq \inf_z f(z, s)$. Using the truncated model (8) in (2) results in the SPS (Stochastic Polyak Stepsize) method*

$$x_{t+1} = x_t - \tau_t g_t, \quad \tau_t := \min\{\alpha_t, \frac{f(x_t, s_t) - C_t}{\|g_t\|^2}\}. \quad (9)$$

*Its stability index is $\delta_t^{\mathrm{SPS}} = \tau_t[1 - \frac{\tau_t}{2\alpha_t}]\|g_t\|^2$. In particular, $\delta_t^{\mathrm{SPS}} \leq \delta_t^{\mathrm{SGD}}$ and*

$$\boxed{\delta_t^{\mathrm{SPS}} \leq \min\{\alpha_t\|g_t\|^2, f(x_t, s_t) - C_t\}.} \quad (10)$$

We can also bound the expected stability index, which is the final quantity that appears in our convergence theorems. Using Lemma A.1 together with (10), we have

$$\mathbb{E}[\delta_t^{\mathrm{SPS}}] \leq \min\{\alpha_t\mathbb{E}\|g_t\|^2, \mathbb{E}[f(x_t) - C_t]\}.$$

The second term in the $\min$-operator can be decomposed as

$$\mathbb{E}[f(x_t) - C_t] = \mathbb{E}[f(x_t) - f(x_\star)]$$
$$+ \underbrace{f(x_\star) - \mathbb{E}_s[\inf_z f(z,s)]}_{=:\sigma^2} + \underbrace{\mathbb{E}_s[\inf_z f(z,s) - C_s]}_{=:\varepsilon_{\mathrm{lb}}}.$$

Here we call $\sigma^2$ the *interpolation constant* and $\varepsilon_{\mathrm{lb}}$ the *lower-bound estimation error*.

> **Takeaway:** We always have $\delta_t^{\mathrm{SPS}} \leq \delta_t^{\mathrm{SGD}}$ (assuming both methods are at the same $x_t$ and $s_t$). The gain in stability for SPS is determined by the interpolation constant, and the lower-bound estimation error.

### 4.3. Square-root model (NGN)

For non-negative loss functions, we can use the NGN (Non-negative Gauss Newton) method proposed by Orvieto & Xiao (2024). They propose to rewrite $f(x) = \mathbb{E}[f(x,s)] = \mathbb{E}[\sqrt{f(x,s)}^2]$ and, for given $s \in \mathcal{S}$, to linearize $y \mapsto \sqrt{f(y,s)}$ around $x$, followed by the square operation. This leads us to the following model: for $g \in \partial f(x,s)$, define

$$f_x(y,s) = \left(\sqrt{f(x,s)} + \frac{1}{2\sqrt{f(x,s)}}\langle g, y-x\rangle\right)^2. \quad (11)$$

This NGN model also enjoys a favorable stability index.

**Lemma 4.2.** *Let $f(x,s) \geq 0$ for all $x \in \mathbb{R}^d$, $s \in \mathcal{S}$. Using model (11) results in the NGN method (Orvieto & Xiao, 2024)*

$$x_{t+1} = x_t - \gamma_t g_t, \quad \gamma_t := \frac{\alpha_t}{1 + \frac{\alpha_t}{2f_t}\|g_t\|^2}. \quad (12)$$

*Its stability index is given by*

$$\boxed{\delta_t^{NGN} = \frac{\gamma_t}{2}\|g_t\|^2 \leq \delta_t^{SGD}.}$$

Not only do we have $\delta_t^{\mathrm{NGN}} \leq \delta_t^{\mathrm{SGD}}$, but further if $\alpha_t \to +\infty$ we have $\gamma_t \to 2\frac{f(x_t,s_t)}{\|g_t\|^2}$. Consequently, $\delta_t^{\mathrm{NGN}}$ does not scale linearly with $\alpha_t$, which suggests that the method should be very stable for large step sizes. Also note the relation to the Polyak step size when $C_t = 0$ (up to a factor of two).

> **Takeaway:** It always holds $\delta_t^{\mathrm{NGN}} \leq \delta_t^{\mathrm{SGD}}$ and further $\delta_t^{\mathrm{NGN}}$ scales sub-linearly with respect to $\alpha_t \to +\infty$.

**On Assumptions (Ã1) and (Ã2).** (Ã1) holds with $\lambda = 0$ due to the convexity of a composition of convex and linear mappings. Regarding (Ã2), we clearly have $f_x(x,s) = f(x,s)$. However, the second part of (Ã2) is less trivial. It is equivalent to the condition that, for some $\rho_s \geq 0$, it holds

$$f(y,s) + \frac{\rho_s}{2}\|y-x\|^2 \geq f(x,s) + \langle g, y-x\rangle + \frac{\langle y-x, g\rangle^2}{4f(x,s)}$$

for all $x, y \in \mathbb{R}^d$, all $s \in \mathcal{S}$ and all $g \in \partial f(x,s)$. Assuming a global upper bound on $\|g\|$ and lower bound on $f(x,s)$ this can be achieved with $\rho_s > \eta_s$. For the convex case, where we require $\rho_s = 0$, the above condition can be satisfied at least on a bounded domain, if $f(\cdot,s)$ is $\alpha$-exp-concave (see Hazan (2019, Def. 4.1). This follows from Hazan (2019, Lem. 4.3), considering that $\frac{1}{2f(x,s)} \leq \frac{1}{2\inf_z f(z,s)}$ and the latter term can be made arbitrarily small by adding a constant to $f(\cdot,s)$.

**Extensions of NGN.** We show in Section D of the Appendix how to extend the key idea of NGN to general composition of maps. In particular, we show that under mild assumptions on these maps, it always holds that the resulting stability index is smaller than the one of SGD.

### 4.4. Exact Model (SPP)

Using the exact model, that is $f_x(y,s) = f(y,s)$, recovers the SPP method. Assume for this part that $\bar{\eta} := \sup_{s\in\mathcal{S}} \eta_s < +\infty$. Then, $f(\cdot,s)$ is $\bar{\eta}$-weakly convex for all $s \in \mathcal{S}$, and update (2) is well-defined for $\alpha_t \in (0, 1/\bar{\eta})$. In particular, (Ã1) is satisfied with $\lambda = \bar{\eta}$, and (Ã2) with $\rho_s = 0$.

**Lemma 4.3.** *Assume that $f(\cdot,s)$ is $\bar{\eta}$-weakly convex and $\alpha_t \in (0, 1/\bar{\eta})$. Then, the stability index for the exact model $f_x(y,s) = f(y,s)$ is bounded by*

$$\boxed{\delta_t^{SPP} \leq \min\{\frac{\alpha_t}{2(1-\alpha_t\bar{\eta})}\|g_t\|^2, f(x_t,s_t) - \inf_{y\in\mathbb{R}^d} f(y,s_t)\}.}$$

*In particular, if $\bar{\eta} = 0$, then $\delta_t^{SPP} \leq \delta_t^{SGD}$ for all $\alpha_t > 0$.*

Using Lemma A.1 reveals the relationship to the interpolation constant $\sigma^2$:

$$\mathbb{E}[\delta_t^{\mathrm{SPP}}] \leq \min\{\frac{\alpha_t}{2(1-\alpha_t\bar{\eta})}\mathbb{E}\|g_t\|^2, \mathbb{E}[f(x_t) - f(x_\star)] + \sigma^2\}.$$

> **Takeaway:** In the convex case, $\delta_t^{\mathrm{SPP}}$ scales sub-linearly with $\alpha_t$. The bound on $\mathbb{E}[\delta_t^{\mathrm{SPP}}]$ is also capped by sub-optimality and the interpolation constant $\sigma^2$.

*Remark* 4.4. Note that for finite-sum problems, where $f(\cdot,s)$ defines a mini-batch loss, the choice of the batch size influences the value of $\sigma^2$ (for example, consider linear regression, where $\inf_z f(z,s) = 0$ as long as the batch size is smaller than the dimension $d$, but $\inf_z f(z,s)$ can be non-zero for large batch sizes). The above shows that the batch size has an impact on the stability of SPP; indeed we observe this in the experiments below, and is further confirmed by the empirical results of Milzarek et al. (2024).

**Verification with PEP.** In contrast to the other methods, for SPP we can only derive an *upper bound* of $\delta_t$. In this

paragraph, we verify whether our bound for $\delta_t^{\text{SPP}}$ is tight in a worst-case sense (for the convex case, $\bar{\eta} = 0$). For this, fix $x \in \mathbb{R}^d$ and let $x_+ = \arg\min_{y \in \mathbb{R}^d} f(y, s) + \frac{1}{2\alpha}\|y - x\|^2$ be the SPP update for fixed $t \in \mathbb{N}$. Denote by $f_s$ the mapping $y \mapsto f(y, s)$. Let $\delta(\alpha) = f_s(x) - \text{env}_{f_s}^\alpha(x)$. Denote by $\mathcal{C}$ the class of lower-bounded convex functions. Assume that $u_\star \in \arg\min_z f_s(z)$. For $G > 0$, and $g \in \partial f_s(x)$, we are interested in the quantity

$$\Lambda(\alpha) := \max_{f_s \in \mathcal{C}} \delta(\alpha) \text{ s.t. } f_s(x) - \min f_s \leq 1, \|g\| \leq G.$$

It is easy to see that $\text{env}_{f_s}^\alpha(x) \searrow \inf f_s$ with $\alpha \to \infty$, and hence $\Lambda(\alpha) \leq f_s(x) - \inf f_s$. However, this bound is not necessarily tight, especially for small $\alpha$.

Therefore, we compute $\Lambda(\alpha)$ using the PEP (Performance Estimation Problem) framework (Drori & Teboulle, 2014; Taylor et al., 2017) implemented in PEPit (Goujaud et al., 2024). Fig. 2b reveals that for the worst-case example the bound $\Lambda(\alpha) \leq \min\{\frac{\alpha}{2}G^2, f_s(x) - \inf f_s(x)\}$ is indeed very tight across all $\alpha > 0$; only in the region where the transition of the minimum happens, we can observe that $\Lambda(\alpha)$ is slightly smaller.

*Remark* 4.5 (Practicality of SPP). SPP is often considered a theoretical method only, as the update in general cannot be solved in closed form. However, Milzarek et al. (2024) show how the update subproblem can be solved in competitive runtime for specific problem structures that naturally appear in statistical learning (generalized linear models). Their technique allows for weakly convex loss functions, mini-batching as well as nonsmooth regularization functions.

### 4.5. Summary

We summarize the main aspects of our theoretical analysis:

**1)** For the question of training stability with respect to large learning rates, the key quantity is $\delta_t$ and how it scales (in expectation) with respect to the step size $\alpha_t$.

**2)** We derive the value of $\delta_t$ for four methods, SGD, SPS, NGN, and SPP. We show that SPS, NGN, and SPP have stability index $\delta_t$ that is smaller or equal to that of SGD. Indeed, for SGD the scaling of $\delta_t$ wrt. $\alpha_t$ is linear, whereas for NGN, SPS and SPP this linear growth is flattened. This is illustrated in Fig. 3. As a by-product of our analysis of $\delta_t$, by plugging our bounds for NGN and SPS into Theorems 3.3 and 3.4, we get the first convergence analysis in the convex and non-smooth setting for these methods (see Section B.1).

**3)** We stress that (weak) convexity of $f(\cdot, s)$ is *not* required to derive the above expressions of $\delta_t$ (with the exception of SPP). Convexity is only required for the argument how $\delta_t$ impacts the suboptimality bound Theorems 3.3 and 3.4. Our extension to weakly convex problems shows that $\delta_t$ is informative of stability also for non-convex problems – and indeed our experiments below confirm that.

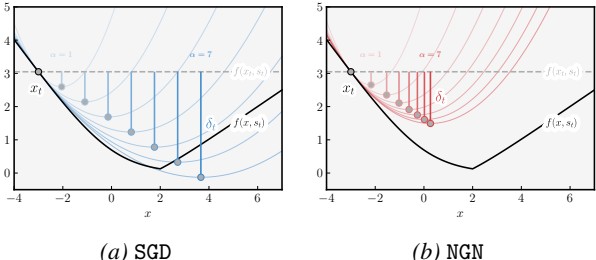

*(a)* SGD    *(b)* NGN

*Figure 3.* Illustration of the stability index $\delta_t$ for SGD (left) and NGN (right), for the function from Fig. 1. Thin colored lines display the objective of the update step (2). For growing $\alpha$ this illustrates how $\delta_t$ grows much slower for NGN compared to SGD.

We state again the key quantity $\delta_t$ for each of the methods we considered.[6] While we focus on these four models, our analysis can be extended to other model-based methods.

$$
\begin{aligned}
\delta_t^{\text{SGD}} &= \tfrac{\alpha_t}{2}\|g_t\|^2, \\
\delta_t^{\text{SPS}} &= \tau_t[1 - \tfrac{\tau_t}{2\alpha_t}]\|g_t\|^2, \quad (\tau_t \text{ from } (9)), \\
\delta_t^{\text{NGN}} &= \tfrac{\gamma_t}{2}\|g_t\|^2, \quad (\gamma_t \text{ from } (12)), \\
\delta_t^{\text{SPP}} &\leq \min\big\{\tfrac{\alpha_t}{2}\|g_t\|^2, f(x_t, s_t) - \inf_{y \in \mathbb{R}^d} f(y, s_t)\big\}.
\end{aligned}
\tag{13}
$$

The above summary on how $\delta_t$ scales with $\alpha_t$ for different methods can be seen even clearer for the special case of quadratic functions where the above expressions simplify further (see Section A.3)

## 5. Experiments

The main objective of the experiment section is to verify whether the theoretical results are reflected in the real-world performance of SGD, SPS, NGN, and SPP for different learning tasks. For simplicity, we always consider the bound $\Omega_T$ from Theorem 3.4. We do this comparison in a **qualitative rather than quantitative** fashion, by looking at final training loss and bound as a function of the step size $\alpha$. Note that for this purpose *training stability* is our main interest rather than generalization properties. Hence, we usually display the training loss. We would like to stress that it is *not* the goal of our experiments to convince the reader of the superior performance of SPS or NGN; their performance has been studied extensively for many deep learning tasks against competitive baselines (Schaipp et al., 2024; Islamov et al., 2025; Schaipp et al., 2023).

**Finite-sum problems.** In all experiments, we consider the objective given as a finite sum over data points, that is $f(x) = \frac{1}{n}\sum_{i=1}^n f_i(x)$ for some $f_i : \mathbb{R}^d \to \mathbb{R}$. For batch size $b \in \mathbb{N}$, we can define $\mathcal{S}$ to be the set of all subsets of cardinality $b \in \mathbb{N}$ of $[n]$ ($s \in \mathcal{S}$ containing the indices of

---

[6]For SPP we state the formula for the convex case, $\bar{\eta} = 0$.

the batch). Now, for $s = \{j_1, \ldots, j_b\}$ define $f(x, s) := \frac{1}{b} \sum_{i=1}^{b} f_{j_i}(x)$. Then, if $s$ follows a uniform distribution, we obtain $\frac{1}{n} \sum_{i=1}^{n} f_i(x) = f(x) = \mathbb{E}_s[f(x, s)]$.

**General setup.** In all experiments, we parametrize the step size/learning rate as $\alpha_t = \alpha \cdot \eta_t$, where we call $\alpha > 0$ the *base learning-rate*, and $(\eta_t)_t \subset \mathbb{R}_{>0}$ is called the *schedule*. Our main interest is the performance of methods as a function of $\alpha$, in particular their stability when $\alpha$ becomes large. By default, we set the schedule $(\eta_t)_t$ to be constant. For SPS, if not explicitly mentioned otherwise, we set the lower bound $C_t = 0$ in every iteration.

### 5.1. non-convex Deep Learning Tasks

We train a ResNet20 on CIFAR10. We run SGD, SPS, and NGN for 20 epochs on a range of base learning-rates $\alpha \in [10^{-2}, 10^2]$. This rather short training horizon will be sufficient to analyze stability wrt. large learning rates: in our experiments, the initial training stage is usually decisive for stability. For SGD, we run two settings:

(1) **(without warmup):** constant learning rate schedule, that is, $\eta_t = 1$ for all $t \in \mathbb{N}$,

(2) **(with warmup):** constant schedule after 100 iterations of linear warmup, from $\eta_1 = 10^{-10}$ to $\eta_{100} = 1$.

We log the gradient norm $\|g_t\|$ and batch loss $f(x_t, s_t)$ in every step, which allows us to track $\{\delta_t^{\text{SGD}}, \delta_t^{\text{SPS}}, \delta_t^{\text{NGN}}\}$. We approximate $\mathbb{E}[\delta_t]$ by averaging over three different seeds (with identical initialization but different data ordering).

*Discussion:* Fig. 4 shows that, despite the problem at hand being non-convex, the bound from Theorem 3.4 closely reflects the range of base learning-rate for which we obtain a good final training loss. In particular, it reflects the fact that SGD starts to degrade for $\alpha \geq 1$, whereas SPS and NGN work well up to $\alpha \approx 10$. For example, the base learning-rate for which the bound of SPS is minimal, matches the one for which the *actual train loss* is smallest. When using SGD with warmup (Fig. 4b), it becomes more stable and works reasonably for roughly one order of magnitude larger $\alpha$. Again, this is closely reflected in the bound as well.

For Fig. 4 we set the unknown parameter $D = 50$ based on $\|x_T - x_1\| \approx 50$.[7] and $T = 20$ epochs. We ablate these choices in Figs. 9 and 10, showing that it does not impact our main observation. Further, Fig. 11 shows that the stability in terms of validation loss is very similar.

Fig. 5 depicts the evolution of $\delta_t$ over time: for SPS and NGN, the values of $\delta_t$ early in training grow much slower com-

---

[7]However, this is highly approximative, as two different methods/ learning rates can result in similar final training loss, but rather distinct final model norm.

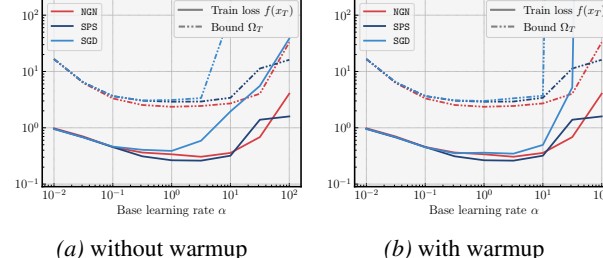

*(a) without warmup*      *(b) with warmup*

*Figure 4.* ResNet20 on CIFAR10: Actual training loss **(solid lines)** and value of the bound $\Omega_T^{\text{last}}$, Theorem 3.4 **(dashed)** with $D = 50$. After $T = 20$ epochs, SPS and NGN are more stable than SGD for large $\alpha$ and achieve a smaller loss. This behavior is (qualitatively) reflected in the bound. **(Right)** Warmup allows SGD to use a larger learning rate. Again, this is reflected in the bound $\Omega_T^{\text{last}}$.

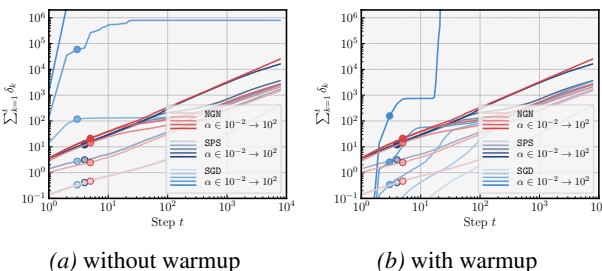

*(a) without warmup*      *(b) with warmup*

*Figure 5.* **(Left)** $\delta_t$ explodes for SGD (without warmup) in the first iterations when $\alpha$ is large. **(Right)** When using warmup over 100 steps, the increase of $\delta_t$ is slowed down.

pared to SGD with the same $\alpha$; this confirms our theoretical finding that $\{\delta_t^{\text{SPS}}, \delta_t^{\text{NGN}}\} \leq \delta_t^{\text{SGD}}$ even though the learning task at hand is non-convex.

### 5.2. Convex Regression Tasks

We also run a set of convex tasks, where the assumptions of the theory are satisfied. First, for ridge regression on synthetic data, we compare SGD, SPP and SPS; we omit NGN here for the sake of clarity of presentation. We further compare SGD, SPS and NGN on linear classification tasks.

**Linear regression.** Assume we have given a feature matrix $\mathbf{A} \in \mathbb{R}^{n \times d}$ and targets $\mathbf{b} \in \mathbb{R}^n$. The objective is given by $f(x) = \frac{1}{2n}\|\mathbf{A}x - \mathbf{b}\|_2^2$. For a batch $s = \{i_1, \ldots, i_b\}$, denote by $\mathbf{A}_s \in \mathbb{R}^{b \times d}$ and $\mathbf{b}_s \in \mathbb{R}^b$ the subset of rows of $\mathbf{A}$ and $\mathbf{b}$ indexed by $s$; hence, we have $f(x, s) = \frac{1}{2b}\|\mathbf{A}_s x - \mathbf{b}_s\|_2^2$. Lemma A.2 in the Appendix derives the SPP update for this specific problem.

*Discussion:* We set $n = 50$, $d = 10$, $b = 5$. The data $\mathbf{A}$, $\mathbf{b}$ is synthetically generated, in a way such that every datapoint can be perfectly fitted (hence we have interpolation, $\sigma^2 = 0$). The schedule is chosen to be constant for all methods.

Fig. 6 shows that SPS and SPP achieve reasonably good performance for values of $\alpha$ up to $10^2$, while SGD starts to fail

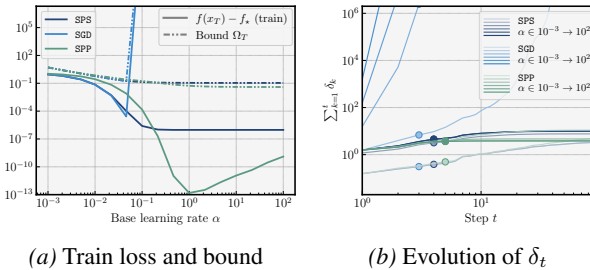

*(a)* Train loss and bound          *(b)* Evolution of $\delta_t$

*Figure 6.* Linear regression: **(Left)** Actual training loss **(solid lines)** and value of the bound $\Omega_T^{\text{last}}$, Theorem 3.4 **(dashed)** with $D = 1$. **(Right)** The values of $\delta_t$ quickly explode for SGD, whereas for SPP and SPS they remain bounded when $\alpha$ becomes large.

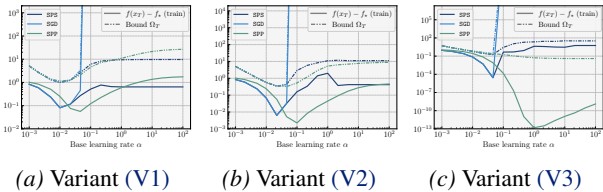

*(a)* Variant (V1)     *(b)* Variant (V2)     *(c)* Variant (V3)

*Figure 7.* Variations for the linear regression problem.

at $\alpha \approx 0.03$. Again, the bound from Theorem 3.4 closely reflects this behavior; it also matches the true performance regarding the fact that SPP "overtakes" SPS for values of $\alpha \geq 1$. Next, we run three problem variations:

(V1) We add noise to the targets **b**. As $n > d$, this results in $f_\star > 0$, while $\inf_z f(z, s) = 0$ still holds (as the batch size is smaller than $d$). Consequently, our theory predicts SPS and SPP to perform worse now.

(V2) We run the noisy setting with larger batch size of $b = 25$ (and 50 epochs instead of 10 to keep the number of steps equal). As $\inf_z f(z, s) > 0$, we don't expect improvement for SPS (as $C_t = 0$), but for SPP to regain advantage over SGD as $\sigma^2$ is smaller than in (V1).

(V3) We again run the noiseless case, but now with $C_t = -2$ for SPS. Here, even though $\sigma^2 = 0$ we expect SPS to loose all advantage compared to SGD.

Fig. 7 shows that for all variants the actual performance is in line with what we would expect according to our theory.

**Multi-class logistic regression.** We run multi-class logistic regression for two datasets (vowel and dna) from LIBSVM (Chang & Lin, 2011). Further dataset details are reported in Section C.

*Discussion:* For both datasets (see Figs. 8 and 12), we again observe that the range of step sizes with good final loss roughly coincides with the range obtained from the theoretical bound (here we set $D = 100$ for vowel and $D = 50$ for dna). In particular, for vowel there is almost

no advantage of SPS over SGD, which is likely due to the lower-bound estimation error. This confirms that even for badly specified lower bounds, SPS does not perform worse than SGD, but it also shows no advantage. NGN is affected by this to a lesser extent.

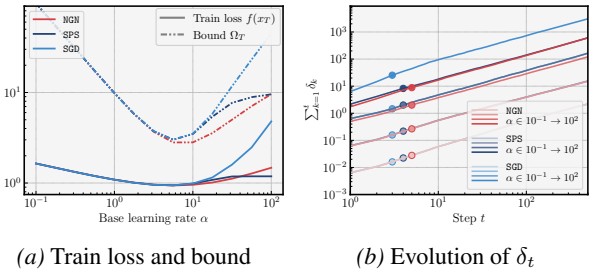

*(a)* Train loss and bound          *(b)* Evolution of $\delta_t$

*Figure 8.* Logistic regression on vowel dataset.

## 6. Conclusion

We have presented a theoretical analysis for step-size stability of several stochastic optimization methods. Within our framework, we prove that adaptive methods such as SPS or NGN are always more stable than SGD in terms of a (weakly) convex convergence bound. Our experiments show a close relationship between the convex bound and the actual training loss, even for non-convex deep learning tasks.

Finally, we want to point out again the main limitations and possible directions for future work. Our current analysis can be converted into convergence rates by applying a standard Lipschitz bound on $\delta_t$ (see Section B.1 for details); however, this is too coarse to capture the advantage of adaptive methods like SPS and NGN and hence not sufficient to show a speedup. Tightening this conversion from bound to convergence rate would be an interesting extension.

On the practical side, an open question is how tracking the stability index $\delta_t$ could be leveraged in practice, for example, to terminate unstable runs early. This paper does not explore such heuristics. Another limitation of our current work is that the analysis is restricted to SGD-based updates, not including momentum, weight decay or preconditioning. For the Polyak step size, it has been shown that it can be combined with all of these (Schaipp et al., 2024), and additionally recent works show that it is beneficial in combination with base optimizers like Muon (Crawshaw et al., 2025) and ScheduleFree (Defazio, 2026). Hence, a direction for future work would be to extend our analysis to other update geometries beyond the model-based framework presented here.

## Acknowledgements

We thank Umut Şimşekli and Francis Bach for comments and feedback. A. Taylor is supported by the European Union (ERC grant CASPER 101162889). Fabian Schaipp is supported by the French government under the management of Agence Nationale de la Recherche as part of the "Investissements d'avenir" program, reference ANR-19-P3IA-0001 (PRAIRIE 3IA Institute), and the European Research Council Starting Grant DYNASTY – 101039676.

## Impact Statement

This paper presents work whose goal is to advance the field of Machine Learning. There are many potential societal consequences of our work, none which we feel must be specifically highlighted here.

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

# A. Auxiliary Lemmas and Missing Proofs

**Lemma A.1.** *Let $X, Y$ be random variables. Then $\mathbb{E}[\min\{X, Y\}] \leq \min\{\mathbb{E}[X], \mathbb{E}[Y]\}$.*

*Proof.* Follows from Jensen's inequality and the fact that $(x, y) \mapsto \min\{x, y\}$ is concave. ☐

**Lemma A.2.** *For $\lambda > 0$, the solution to*

$$x_{t+1} = \arg\min_{y \in \mathbb{R}^d} \frac{1}{2b}\|\mathbf{A}_{s_t} y - \mathbf{b}_{s_t}\|^2 + \frac{\lambda}{2}\|y\|^2 + \frac{1}{2\alpha_t}\|y - x_t\|^2$$

*is given by the solution to the linear system*

$$\left[\tfrac{1}{b}\mathbf{A}_{s_t}^T \mathbf{A}_{s_t} + (\lambda + \alpha_t^{-1})\mathbf{Id}\right]x_{t+1} = \alpha_t^{-1}x_t + \tfrac{1}{b}\mathbf{A}_{s_t}^T \mathbf{b}_{s_t}.$$

*Proof.* Follows from the necessary and sufficient optimality conditions. ☐

## A.1. Proofs for Section 3.2

*Proof of Lemma 3.5.* Let us define $\Psi_t(x) = f_{x_t}(x, s_t) + \frac{1}{2\alpha_t}\|x - x_t\|^2$. Then, due to (Ã1), $\Psi_t$ is $(\frac{1}{\alpha_t} - \lambda)$-strongly convex. Further, $x_{t+1}$ is a minimizer of $\Psi_t$. Therefore, for any $u \in \mathbb{R}^d$ it holds

$$\Psi_t(x_{t+1}) + \frac{1 - \lambda\alpha_t}{2\alpha_t}\|u - x_{t+1}\|^2 \leq \Psi_t(u).$$

Plugging in $u = \hat{x}_t$ and the definition of $\Psi_t$, we get

$$\frac{1 - \lambda\alpha_t}{2\alpha_t}\|x_{t+1} - \hat{x}_t\|^2 \leq \frac{1}{2\alpha_t}\|\hat{x}_t - x_t\|^2 - \frac{1}{2\alpha_t}\|x_{t+1} - x_t\|^2 - [f_{x_t}(x_{t+1}, s_t) - f_{x_t}(\hat{x}_t, s_t)]. \tag{14}$$

Similar to the proof of Lemma 3.2, from (Ã1) and (Ã2) we get

$$f_{x_t}(x_{t+1}, s_t) - f_{x_t}(\hat{x}_t, s_t) = \underbrace{f_{x_t}(x_{t+1}, s_t) - f(x_t, s_t)}_{=-\delta_t - \frac{1}{2\alpha_t}\|x_{t+1} - x_t\|^2} + f(x_t, s_t) - f_{x_t}(\hat{x}_t, s_t)$$

$$\geq f(x_t, s_t) - f(\hat{x}_t, s_t) - \frac{\rho_{s_t}}{2}\|x_t - \hat{x}_t\|^2 - \delta_t - \frac{1}{2\alpha_t}\|x_{t+1} - x_t\|^2.$$

Plugging this back in (14), multiplying by $2\alpha_t$, and taking conditional expectation, we get

$$(1 - \lambda\alpha_t)\mathbb{E}_t\|x_{t+1} - \hat{x}_t\|^2 \leq (1 + \alpha_t\rho)\|\hat{x}_t - x_t\|^2 - 2\alpha_t[f(x_t) - f(\hat{x}_t)] + 2\alpha_t\mathbb{E}_t[\delta_t].$$

Since (Ã1)-(Ã2) imply weak convexity of $f$ and $\bar{\rho} > \rho + \lambda$, by definition of the proximal operator we have

$$f(x_t) - f(\hat{x}_t) \geq \frac{\bar{\rho}}{2}\|x_t - \hat{x}_t\|^2.$$

Therefore, we get

$$(1 - \lambda\alpha_t)\mathbb{E}_t\|x_{t+1} - \hat{x}_t\|^2 \leq \big(1 - \alpha_t(\bar{\rho} - \rho)\big)\|x_t - \hat{x}_t\|^2 + 2\alpha_t\mathbb{E}_t[\delta_t].$$

Multiplying by $(1 - \lambda\alpha_t)^{-1}$ gives the final result. ☐

*Proof of Theorem 3.6.* Denote again $\hat{x}_t = \text{prox}_{(1/\bar{\rho})f}(x_t)$. By definition of the Moreau envelope and applying Lemma 3.5, we have

$$\mathbb{E}_t[\text{env}_f^{1/\bar{\rho}}(x_{t+1})] \leq \mathbb{E}_t[f(\hat{x}_t) + \frac{\bar{\rho}}{2}\|\hat{x}_t - x_{t+1}\|^2]$$

$$\leq f(\hat{x}_t) + \frac{\bar{\rho}}{2}\Big[\|x_t - \hat{x}_t\|^2 - \frac{\alpha_t(\bar{\rho} - \rho - \lambda)}{1 - \alpha_t\lambda}\|x_t - \hat{x}_t\|^2 + \frac{2\alpha_t}{1 - \alpha_t\lambda}\mathbb{E}_t[\delta_t]\Big].$$

For the third term, we use that $\|x_t - \hat{x}_t\|^2 = (\bar{\rho})^{-2}\|\nabla\text{env}_f^{1/\bar{\rho}}(x_t)\|^2$ (Davis & Drusvyatskiy, 2019, Lem. 2.2). Thus, we get

$$\mathbb{E}_t[\text{env}_f^{1/\bar{\rho}}(x_{t+1})] \leq \text{env}_f^{1/\bar{\rho}}(x_t) - \frac{\alpha_t(\bar{\rho} - \rho - \lambda)}{2\bar{\rho}(1 - \alpha_t\lambda)}\|\nabla\text{env}_f^{1/\bar{\rho}}(x_t)\|^2 + \frac{\alpha_t\bar{\rho}}{1 - \alpha_t\lambda}\mathbb{E}_t[\delta_t].$$

Applying total expectation, and summing over $t = k, \ldots, T$, we obtain

$$\sum_{t=k}^{T} \frac{\alpha_t}{1 - \alpha_t\lambda}\mathbb{E}\|\nabla\text{env}_f^{1/\bar{\rho}}(x_t)\|^2 \leq \frac{2\bar{\rho}}{\bar{\rho} - \rho - \lambda}\left[\mathbb{E}[\text{env}_f^{1/\bar{\rho}}(x_k)] - \mathbb{E}[\text{env}_f^{1/\bar{\rho}}(x_{T+1})]\right] + \frac{2\bar{\rho}^2}{\bar{\rho} - \rho - \lambda}\sum_{t=k}^{T}\frac{\alpha_t}{1 - \alpha_t\lambda}\mathbb{E}[\delta_t].$$

$\square$

## A.2. Proofs for Section 4

For the following proofs, recall the notation from (6), that is

$$\Psi_t = f_{x_t}(x_{t+1}, s_t) + \frac{1}{2\alpha_t}\|x_{t+1} - x_t\|^2.$$

*Proof of Lemma 4.1.* Denote $C_t := C_{s_t}$. Due to Schaipp et al. (2023, Lem. 9) it holds $x_{t+1} = x_t - \tau_t g_t$ with

$$\Psi_t = f(x_t, s_t) - \tau_t\|g_t\|^2 + \frac{\tau_t^2}{2\alpha_t}\|g_t\|^2$$

$$= f(x_t, s_t) - \tau_t[1 - \frac{\tau_t}{2\alpha_t}]\|g_t\|^2.$$

Since $\Psi_t = f(x_t, s_t) - \delta_t^{\text{SPS}}$ we have from the above that $\delta_t^{\text{SPS}} = \tau_t[1 - \frac{\tau_t}{2\alpha_t}]\|g_t\|^2$. Given that the function $\tau \mapsto \tau(1 - \frac{\tau}{2\alpha})$ is maximized at $\hat{\tau} = \alpha$, we conclude that $\delta_t^{\text{SPS}} \leq \frac{\alpha_t}{2}\|g_t\|^2 = \delta_t^{\text{SGD}}$ (assuming both $\delta_t$ would be measured at the same iterate $x_t$ and batch $s_t$).

Now, we treat the two cases separately:

- If $\tau_t = \frac{f(x_t, s_t) - C_t}{\|g_t\|^2}$, we have $1 - \frac{\tau_t}{2\alpha_t} \leq 1$ and hence $\delta_t^{\text{SPS}} \leq \tau_t\|g_t\|^2 = f(x_t, s_t) - C_t \leq \alpha_t\|g_t\|^2$.

- In the other case, if $\tau_t = \alpha_t$ then $\delta_t^{\text{SPS}} = \tau_t[1 - \frac{\tau_t}{2\alpha_t}]\|g_t\|^2 = \frac{\alpha_t}{2}\|g_t\|^2 \leq \frac{1}{2}[f(x_t, s_t) - C_t]$.

Combining both cases we can upper-bound

$$\delta_t^{\text{SPS}} \leq \min\{\alpha_t\|g_t\|^2, f(x_t, s_t) - C_t\}. \tag{15}$$

Consequently $\delta_t^{\text{SPS}}$ scales sub-linearly with $\alpha_t$. $\square$

*Proof of Lemma 4.2.* For $x_t \in \mathbb{R}^d$ and $g_t \in \partial f(x_t, s_t)$, denote $f_t := f(x_t, s_t)$. Orvieto & Xiao (2024) show that update (2) with model (11) is given by (12). Plugging into (11), we get

$$f_{x_t}(x_{t+1}, s_t) = f_t - \gamma_t\|g_t\|^2 + \frac{1}{4f_t}\gamma_t^2\|g_t\|^4,$$

and $\frac{1}{2\alpha_t}\|x_{t+1} - x_t\|^2 = \frac{1}{2\alpha_t}\gamma_t^2\|g_t\|^2$. Hence,

$$\Psi_t = f_t - \gamma_t\|g_t\|^2 + \gamma_t^2\underbrace{\left[\frac{1}{4f_t}\|g_t\|^4 + \frac{1}{2\alpha_t}\|g_t\|^2\right]}_{=\frac{\|g_t\|^2}{2\alpha_t}(1+\frac{\alpha_t\|g_t\|^2}{2f_t})=\frac{\|g_t\|^2}{2\gamma_t}}$$

$$= f_t - \frac{\gamma_t}{2}\|g_t\|^2.$$

Altogether, we obtain $\delta_t^{\text{NGN}} = f_t - \Psi_t = \frac{\gamma_t}{2}\|g_t\|^2$. From $\gamma_t \leq \alpha_t$ it follows that $\delta_t^{\text{NGN}} \leq \delta_t^{\text{SGD}}$. $\square$

*Proof of Lemma 4.3.* We have $\Psi_t - f(x_t, s_t) = f(x_{t+1}, s_t) - f(x_t, s_t) + \frac{1}{2\alpha_t}\|x_{t+1} - x_t\|^2$. For any $\alpha_t \in (0, 1/\bar{\eta})$, it holds

$$\Psi_t = \min_{y \in \mathbb{R}^d} f(y, s_t) + \frac{1}{2\alpha_t}\|y - x_t\|^2 \geq \min_{y \in \mathbb{R}^d} f(y, s_t).$$

Hence, we have

$$\Psi_t - f(x_t, s_t) \geq \inf_{y \in \mathbb{R}^d} f(y, s_t) - f(x_t, s_t). \tag{16}$$

On the other hand, due to $\bar{\eta}$-weak convexity of $f(\cdot, s)$ we have

$$f(x_{t+1}, s_t) \geq f(x_t, s_t) + \langle g_t, x_{t+1} - x_t \rangle - \frac{\bar{\eta}}{2}\|x_{t+1} - x_t\|^2$$

$$\geq f(x_t, s_t) - \frac{\varepsilon}{2}\|g_t\|^2 - (\frac{1}{2\varepsilon} + \frac{\bar{\eta}}{2})\|x_{t+1} - x_t\|^2,$$

where we use Young's inequality with $\varepsilon > 0$ in the second step. Setting $\varepsilon = \frac{\alpha_t}{1 - \alpha_t\bar{\eta}} > 0$, we get

$$\Psi_t - f(x_t, s_t) \geq -\frac{\alpha_t}{2(1 - \alpha_t\bar{\eta})}\|g_t\|^2. \tag{17}$$

Combining (16) and (17) , we have

$$\Psi_t - f(x_t, s_t)$$

$$\geq \max\{-\frac{\alpha_t}{2(1 - \alpha_t\bar{\eta})}\|g_t\|^2, \inf_{y \in \mathbb{R}^d} f(y, s_t) - f(x_t, s_t)\}$$

$$= -\min\{\frac{\alpha_t}{2(1 - \alpha_t\bar{\eta})}\|g_t\|^2, f(x_t, s_t) - \inf_{y \in \mathbb{R}^d} f(y, s_t)\}.$$

From (6), this implies $\delta_t^{\text{SPP}} \leq \min\{\frac{\alpha_t}{2(1 - \alpha_t\bar{\eta})}\|g_t\|^2, f(x_t, s_t) - \inf_{y \in \mathbb{R}^d} f(y, s_t)\}$. $\qquad\square$

## A.3. Quadratics as a Special Case

Consider the class of functions $f(x, s) := \frac{s^2}{2}\|x - x_\star\|^2$ for $s \in \mathbb{R}$ and some fixed $x_\star \in \mathbb{R}^d$. This represents a linear regression problem with interpolation condition, i.e. $x_\star = \arg\min_x f(x, s)$ for all $s \in \mathcal{S}$ and hence $\sigma^2 = 0$. In this case, the formulae for $\delta_t^{\{\text{SGD,SPS,NGN,SPP}\}}$ simplify a lot, and therefore this serves as a good illustration of the different stability behaviors.

First, note that $\|\nabla f(x, s)\|^2 = 2s^2 f(x, s)$ for any $x \in \mathbb{R}^d$, $s \in \mathbb{R}$. From now on, assume that $x_t \in \mathbb{R}^d$ is the (fixed) current iterate, and denote $f_t := f(x_t, s_t) = \frac{\|g_t\|^2}{2s_t^2}$. For SPS, we can clearly use $C_t = 0$ in this case, and obtain $\tau_t = \min\{\alpha_t, \frac{f_t}{\|g_t\|^2}\} = \min\{\alpha_t, \frac{1}{2s_t^2}\}$. Plugging into the exact formula for $\delta_t^{\text{SPS}}$ from Lemma 4.1, and doing a simple case distinction, we obtain $\delta_t^{\text{SPS}} \leq \min\{\frac{\alpha_t}{2}, \frac{1}{2s_t^2}\}\|g_t\|^2$. Note that this bound is relatively tight, and can be made exact at the cost of a less compact expression. For NGN, we have $\gamma_t = \frac{\alpha_t}{1 + \alpha_t s_t^2} \leq \frac{1}{s_t^2}$ for all $\alpha_t > 0$. For SPP, we can compute $x_{t+1} = \frac{1}{1 + \alpha_t s_t^2}[x_t + \alpha_t s_t^2 x_\star]$ and $\frac{1}{2\alpha_t}\|x_{t+1} - x_t\|^2 = \frac{1}{2}\frac{\alpha_t s_t^4}{(1 + \alpha_t s_t^2)^2}\|x_t - x_\star\|^2 = \frac{\alpha_t s_t^2}{(1 + \alpha_t s_t^2)^2}f_t$. This leads to $\delta_t^{\text{SPP}} = \frac{\alpha_t s_t^2}{1 + \alpha_t s_t^2}f_t$.

Altogether, we obtain the following stability indices:

$$\delta_t^{\text{SGD}} = \frac{\alpha_t}{2}\|g_t\|^2,$$

$$\delta_t^{\text{SPS}} \leq \min\{\frac{\alpha_t}{2}, \frac{1}{2s_t^2}\}\|g_t\|^2,$$

$$\delta_t^{\text{NGN}} = \frac{\alpha_t}{2(1 + \alpha_t s_t^2)}\|g_t\|^2,$$

$$\delta_t^{\text{SPP}} = \frac{\alpha_t}{2}\frac{1}{1 + \alpha_t s_t^2}\|g_t\|^2.$$

In conclusion, only $\delta_t^{\text{SGD}}$ scales linearly with $\alpha_t$, whereas for all other methods, their stability index does not grow linearly when $\alpha_t \to \infty$.

# B. Discussion of Related Work

## B.1. Comparison of Convergence Theorems to the Literature

Here we consider our resulting convergence results Theorems 3.3 and 3.4, when instantiated with each model, and compare to what is known in the literature. For the linear model and the resulting SGD method, we recover the standard rates as pointed out in Section 4.1.

For (SPS) and (NGN) we have shown in Lemma 4.1 and Lemma 4.2 respectively, that these methods have a stability constant $\delta_t$ that is less or equal than that of of SGD, that is

$$\delta_t \;\leq\; \delta_t^{\text{SGD}} = \frac{\alpha_t}{2}\|g_t\|^2. \tag{18}$$

Using this bound in Theorems 3.3 and 3.4 we immediately arrive at a convergence result for SPS and NGN. However, this convergence result will be pessimistic, since we bound away the possible stability benefits of SPS and NGN when using (18).

Using the pessimistic upper bound (18) in Theorem 3.3 with a constant step size $\alpha_t \equiv \alpha$ we arrive at the convergence rate

$$\bar{f}_T - f(x_\star) \leq \frac{D^2}{2T\alpha} + \alpha \frac{\sum_{t=1}^{T}\mathbb{E}[\|g_t\|^2]}{2T}. \tag{19}$$

By further assuming the loss is Lipschitz continuous with $\|g_t\| \leq G$ and a step size of $\alpha = \frac{1}{\sqrt{T}}$ this gives a $\mathcal{O}(1/\sqrt{T})$ complexity. This is exactly the rate of SGD in the convex and Lipschitz setting, see for the an early such proof Shamir & Zhang (2013) and Theorem 9.7 in Garrigos & Gower (2024) for a didactic reference.

For SPS (introduced by Loizou et al. (2021) and called SPS$_{\text{max}}$ therein), as far as we are aware, the only result for convex and Lipschitz losses also requires interpolation (Loizou et al., 2021, Thm. C.1), which gives a $\mathcal{O}(1/\sqrt{T})$ complexity.

As for the NGN method, there currently exists only an analysis of the method for the Lipschitz-smooth setting. Thus (19) is the first complexity for NGN in the convex non-smooth setting.

## B.2. Additional Remarks on Related Work

The theory by Loizou et al. (2021) suggests that SPS can achieve the standard convergence rates without knowledge of the Lipschitz(-smoothness) constant. More concretely, Loizou et al. (2021, Thm. 3.4) shows that if $\alpha_t = \alpha$ for all $t \in \mathbb{N}$, and if each $f(\cdot, s)$ is convex and $L_s$-smooth, we have

$$\mathbb{E}[f(\frac{1}{T}\sum_{t=1}^{T} x_t) - f(x_\star)] \leq \frac{\|x_1 - x_\star\|^2}{\rho T} + \frac{2\sigma^2 \alpha}{\rho}.$$

Here $\rho := \min\{\frac{1}{2L_{\max}}, \alpha\}$ and $L_{\max} = \max_s L_s$. However, this theory can not explain why SPS is less sensitive to the choice of $\alpha$: for $\alpha \geq \frac{1}{2L_{\max}}$, the bound is increasing in $\alpha$ linearly (as $\rho = \frac{1}{2L_{\max}}$). Moreover, in the smooth case, it holds

$$\frac{f(x_t, s_t) - \inf_z f(z, s_t)}{\|g_t\|^2} \geq \frac{1}{2L_{s_t}} \geq \frac{1}{2L_{\max}}.$$

For ResNet20 on CIFAR10, the term on the left is around $10^{-1}$ in early training, suggesting that $2L_{\max} \geq 10^1$ and hence the bound will be increasing for $\alpha \geq 10^{-1}$. But in practice, SPS achieves the best performance for much larger values of $\alpha$ (in between 1 and 10, see Fig. 4).

For stochastic proximal point (SPP), several analyses show that convergence can be proven for any step size, for example Tovmasyan et al. (2025, Thm. 5.3, 6.4); however, these analyses derive rates for SPP under various set of (additional) assumptions, such as smoothness, strong convexity, or interpolation; our analysis on the other hand focuses on the one-step stability index $\delta_t$, which allows to compare stability across methods.

Orabona & D'Orazio (2026) show negative results for stability and convergence of the Polyak step size; however, their results are not in conflict with ours, as their examples involve suboptimal estimates for $C_s$ and require large values for $\alpha_t$ (some results only hold for $\alpha_t = +\infty$). This is in line with our analysis, as the stability index becomes worse when both $\varepsilon_{\text{lb}}$ and $\alpha_t$ grow simultaneously.

## C. Experiments: Supplementary Material

**Dataset generation for linear regression.** The data generation procedure was adapted from Loizou et al. (2021) and the reference therein: we first generate a matrix $\mathbf{A} \in \mathbb{R}^{n \times d}$ with entries from a standard normal. We add one to all entries to introduce a bias. We then scale each column with a factor drawn from $\mathcal{N}(0, 1)$ multiplied by ten. We now sparsify $\mathbf{A}$ by setting each entry to zero with probability $1 - 30\frac{\ln(n)}{n}$. We finally scale each column to have norm 10 (this step affects mainly the scaling of the step size).

We then generate a vector $\hat{x} \in \mathbb{R}^d$ with entries from a standard normal, re-scaled such that $\|\hat{x}\| = 1$. We generate $\mathbf{b} = \mathbf{A}\hat{x}$; in the noisy setting, we add a vector with entries from a standard normal to $\mathbf{A}$.

## D. Generalizations of the NGN Step Size

The NGN model (11) is constructed as square-root $\rightarrow$ linearize $\rightarrow$ square. This idea can be generalized to other functions on the non-negative halfspace, and their respective inverse function. Below, we derive the model-based update when using instead $\log \rightarrow$ linearize $\rightarrow \exp$. We will see that the resulting update is related to the Lambert–$W$ function.

**General maps.** Let $U \subseteq \mathbb{R}_{\geq 0}$ and $\phi : U \rightarrow \mathbb{R}$ and $\psi : \mathbb{R} \rightarrow \mathbb{R}$ satisfy the following:

(B1) For any $x \in U$ we have $\psi(\phi(x)) = x$.

(B2) $\psi$ is differentiable and convex on $\mathbb{R}$.

(B3) $\phi$ is differentiable on $U$ with $\phi'(x) \neq 0$ for all $x \in U$.

Second, (B1) implies that $\phi$ is injective. Therefore, $\phi$ must be either monotonically increasing or decreasing. If $\phi$ is concave, then its first derivative is non-increasing, and hence $\phi'(0) > 0$ is sufficient for (B3) to hold with $U = \mathbb{R}_{\geq 0}$.

We will usually choose $U = \mathbb{R}_{>0}$ or $U = \mathbb{R}_{\geq 0}$ depending on the loss function being strictly positive or non-negative. (In the latter case, assume that there exists a continuous extension of $\phi'$ on $\mathbb{R}_{\geq 0}$ with $\phi'(0) \neq 0$.)

Then, for $g \in \partial f(x, s)$, define the model

$$f_x(y, s) = \psi\big(\phi(f(x, s)) + \phi'(f(x, s))\langle g, y - x \rangle\big). \tag{20}$$

First, note that Assumption (A1) holds due to convexity of $\psi$. Denote $f_t := f(x_t, s_t)$, and let $u_t = \phi'(f_t)$. It holds

$$x_{t+1} = \underset{y \in \mathbb{R}^d}{\arg\min}\, f_{x_t}(y, s_t) + \frac{1}{2\alpha_t}\|y - x_t\|^2$$

if and only if

$$x_{t+1} = x_t - \alpha_t u_t v_t g_t, \quad v_t := \psi'\big(\phi(f_t) + \phi'(f_t)\langle g_t, x_{t+1} - x_t \rangle\big).$$

Therefore, we get the condition

$$v_t = \psi'\big(\phi(f_t) - \alpha_t v_t u_t^2 \|g_t\|^2\big).$$

We can compute

$$f_{x_t}(x_{t+1}, s_t) = \psi\big(\phi(f_t) - \alpha_t v_t u_t^2 \|g_t\|^2\big),$$

and hence

$$
\begin{aligned}
\delta_t &= f_t - \frac{1}{2\alpha_t}\|x_{t+1} - x_t\|^2 - f_{x_t}(x_{t+1}, s_t) \\
&= f_t - \frac{1}{2}\alpha_t u_t^2 v_t^2 \|g_t\|^2 - \psi\big(\phi(f_t) - \alpha_t v_t u_t^2 \|g_t\|^2\big).
\end{aligned}
\tag{21}
$$

Let us focus on the last term: using convexity, we have $\psi(a + b) \geq \psi(a) + \psi'(a)b$, and hence

$$\psi\big(\phi(f_t) - \alpha_t v_t u_t^2 \|g_t\|^2\big) \geq \psi(\phi(f_t)) - \psi'(\phi(f_t))\alpha_t v_t u_t^2 \|g_t\|^2.$$

Now, from (B1) and the chain rule it follows that $1 = \psi'(\phi(x))\phi'(x)$ holds for all $x \in U$. Due to (B3), it follows $\psi'(\phi(f_t)) = 1/\phi'(f_t) = 1/u_t$. Together with (B1), we get

$$\psi\big(\phi(f_t) - \alpha_t v_t u_t^2 \|g_t\|^2\big) \geq f_t - \alpha_t v_t u_t \|g_t\|^2.$$

Plugging back into (21), we have

$$\delta_t \leq f_t - \frac{1}{2}\alpha_t u_t^2 v_t^2 \|g_t\|^2 - \big(f_t - \alpha_t v_t u_t \|g_t\|^2\big)$$
$$= \frac{\alpha_t}{2}\|g_t\|^2[2v_t u_t - u_t^2 v_t^2] \leq \frac{\alpha_t}{2}\|g_t\|^2,$$

where the last step uses $1 - 2u_t v_t + u_t^2 v_t^2 = (1 - u_t v_t)^2 \geq 0$.

**Using log and exponential.** Consider (20) with $\psi = \exp$ and $\phi = \log$, that is

$$f_x(y, s) = \exp\Big(\log f(x, s) + \frac{1}{f(x, s)}\langle g, y - x\rangle\Big). \tag{22}$$

The model-based update (2) has the necessary and sufficient optimality condition

$$0 = \exp\big(\tfrac{1}{f(x_t,s_t)}\langle g_t, y - x_t\rangle\big)g_t + \frac{1}{\alpha_t}(y - x_t).$$

Using the Ansatz $y = x_t - \gamma_t g_t$, we obtain

$$\gamma_t = \alpha_t \exp\Big(-\gamma_t \frac{\|g_t\|^2}{f(x_t, s_t)}\Big). \tag{23}$$

Let $W_0$ be the principal branch of the Lambert–$W$ function, which is defined as the inverse function of $z \mapsto ze^z$. Multiplying (23) by $\frac{\|g_t\|^2}{f(x_t,s_t)}\exp\big(\gamma_t \frac{\|g_t\|^2}{f(x_t,s_t)}\big)$, we can see that (23) has the solution

$$\gamma_t = \frac{f(x_t, s_t)}{\|g_t\|^2}W_0\big(\alpha_t \frac{\|g_t\|^2}{f(x_t, s_t)}\big) \quad \text{if } g_t \neq 0 \text{ and else } \gamma_t = \alpha_t. \tag{24}$$

The solution lies on the principal branch as $\alpha_t \frac{\|g_t\|^2}{f(x_t,s_t)} > 0$.

Denote again $f_t = f(x_t, s_t)$. Further, we have $u_t = \frac{1}{f_t}$ and $\alpha_t u_t v_t = \gamma_t$. Hence, $v_t = \exp(\log f_t - \frac{\gamma_t}{f_t}\|g_t\|^2) = f_t \exp(-\frac{\gamma_t}{f_t}\|g_t\|^2) = \frac{f_t \gamma_t}{\alpha_t}$. We can plug this into (21), we get

$$\delta_t = f_t - \frac{\gamma_t^2}{2\alpha_t}\|g_t\|^2 - \exp(\log f_t - \frac{\gamma_t}{f_t}\|g_t\|^2) = f_t - \frac{\gamma_t^2}{2\alpha_t}\|g_t\|^2 - f_t \exp(-\frac{\gamma_t}{f_t}\|g_t\|^2)$$
$$= f_t - \frac{\gamma_t^2}{2\alpha_t}\|g_t\|^2 - \frac{f_t \gamma_t}{\alpha_t}.$$

To the best of our knowledge, the adaptive step size (24) is novel in the literature. In preliminary experiments, we observe that (24) is also very stable with respect to the choice of $\alpha_t$. However, we do not further investigate the behavior of this new step size here, as it would distract from the focus of this paper. Analyzing (24) might be interesting future work.

**Datasets for logistic regression.** We use the scaled versions of the datasets when available. We always use $20\%$ of the `LIBSVM` train split as validation set. Further, we drop the last batch if it is smaller than the batch size. This explains the difference between the sample size reported here and the one on the `LIBSVM` website.

We run 10 epochs for `dna` and 20 epochs for `vowel`.

*Table 1.* Dataset information for multi-class logistic regression experiments.

| Name | Dimension $d$ | Samples $n$ | Class labels $C$ | Batch size $b$ |
|------|------|------|------|------|
| vowel | 10 | 416 | 11 | 16 |
| dna | 180 | 1600 | 3 | 16 |

# E. Additional Plots and Experiments

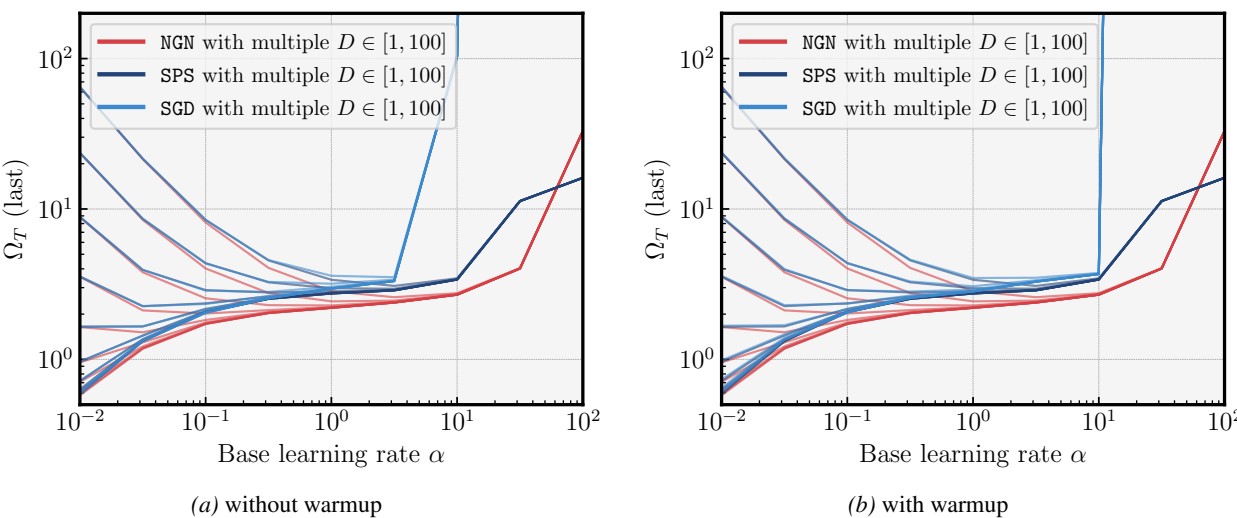

*(a)* without warmup

*(b)* with warmup

*Figure 9.* `ResNet20` on `CIFAR10`: Bound $\Omega_T$ from Theorem 3.4 for multiple values of $D \in [10^0, 10^2]$. The advantage of `SPS` and `NGN` over `SGD` for large $\alpha$ is reflected in the bound independent of the value of $D$.

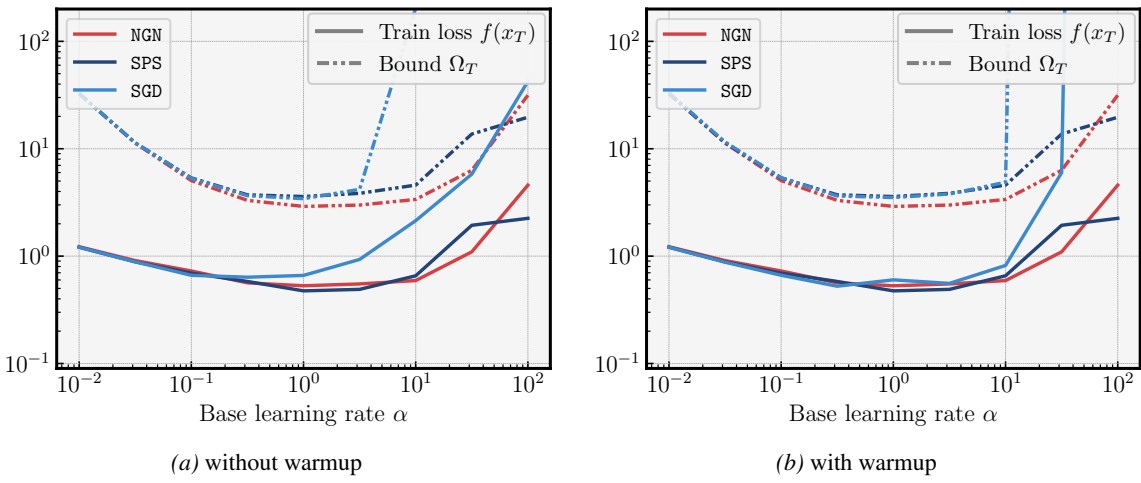

*(a)* without warmup

*(b)* with warmup

*Figure 10.* `ResNet20` on `CIFAR10`: Same as Fig. 4, but computed after 10 instead of 20 epochs.

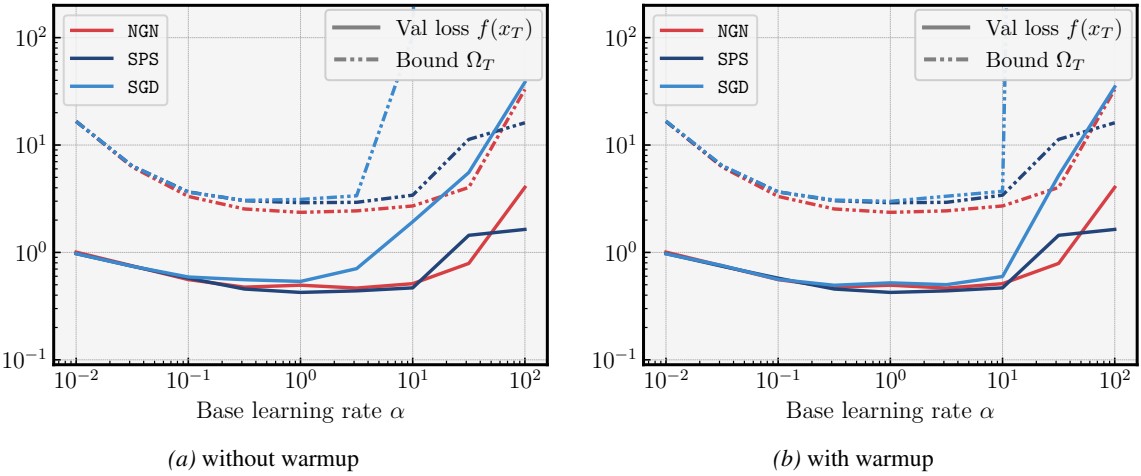

*(a)* without warmup          *(b)* with warmup

*Figure 11.* `ResNet20` on `CIFAR10`: Same as Fig. 4, but plotting validation loss instead of train loss.

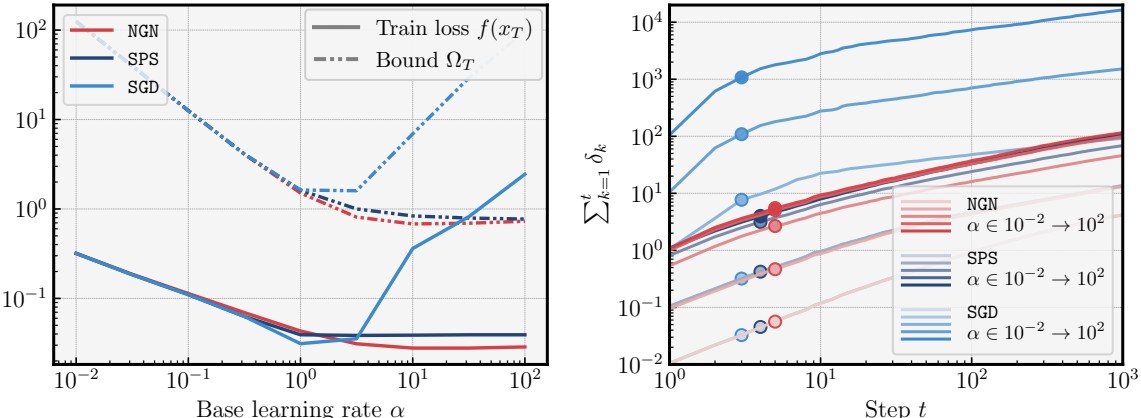

*Figure 12.* Same as Fig. 8 but for logistic regression on `dna` dataset.

