# OpenReview forum: "Step-Size Stability in Stochastic Optimization: A Theoretical Perspective"
_ICML.cc/2026/Conference — ICML 2026 regular_

### Official Review · Reviewer_vs6x · 2026-03-02

**Soundness:** 1
**Presentation:** 2
**Significance:** 1
**Originality:** 1
**Overall Recommendation:** 3
**Confidence:** 4

**Summary:**

This paper aims to provide a theoretical analysis of stochastic optimization algorithms, like stochastic gradient descent (SGD), stochastic polyak step size (SPS), non-negative Gauss-Newton steps (NGN) and stochastic proximal point (SPP) in terms of their sensitivity with respect to the step size. Specifically, this work proposes a quantity called **stability index** for determining how the performance of these algorithms change as a function of step size. Moreover, they run experiments to demonstrate that their theoretical bounds are tight.

**Compliance With Llm Reviewing Policy:**

Affirmed.

**Final Justification:**

I increased my score as I misunderstood the Figure 7. However, the practical applicability and novelty in proof techniques is still questionable. Therefore, I updated my score to 3.

**Key Questions For Authors:**

- I do not understand why there is always a spike (in Figure 4, 7) in the theoretical bounds for SGD. Given the explicit form of $\delta_t$ and constant step size $\alpha$, the authors can provide the closed form expression for $\Omega$ that they use for plotting in Figure 4, 7. That will help the reader understand the behavior much better.

**Limitations:**

- There is no practical application of the theory derived.

- There is no novelty in theory. All results follow directly from previous works.

Also check the weaknesses.

**Strengths And Weaknesses:**

Weaknesses:
- The authors also implicitly assumes that there exist a particular solution $x_*$ such that $f(x_*) = f_*$. Therefore, the theory is not true for general convex functions. For example, $f(x) = e^x$ with minimum at $0$ has no solution $x_*$ such that $f(x_*) = 0$.

- The derivation of NGN step size in Section 4.3 assumes that the function $f$ is always. This puts extra structure on the function and therefore, the comparison of NGN step size with other step sizes’ results are unfair.
As far as I understand, the plots basically show that the stochastic algorithms like SPP, NGN and SPS allow for larger step size than SGD. But why is that important? One still needs to tune the step sizes to get optimal performance for any of these algorithms and the optimal performances for all the algorithms are almost similar (from the Figures in experiments).

- The analysis is not optimal. The authors derive the bound in Theorem 3.3 and 3.4 which holds for any generic $\alpha_t$. They do not exploit the structure of step sizes for each of these algorithms. Moreover, there is no discussion or comparison about the rates that they get after inserting the computed $\delta_t$ for each of these algorithms into the bounds $\Omega_T$. Do they match the optimal bounds derived in the literature?

- Moreover, the bounds can not explain the behavior in convex problems. Check Figure 7.

- In the second point of Summary and Contributions of the paper, the author mention, SPS and NGN are more **stable** than SGD. What does the author mean by that? Stability is never quantitatively defined in the paper.

- As the authors also agree in the limitation section, the practical importance of this paper is questionable.
Moreover, there is no contribution of this paper in terms of theory. Theorems 3.3 and 3.4 are easy to derive. Overall, the contribution of the paper is questionable.

---

> ### Author Rebuttal · Authors · 2026-03-30
>
> We thank the reviewer for their time assessing our submission. We hope that after reviewing the rebuttal, we can convince the reviewer that several weaknesses mentioned are in fact not fundamentally limiting our contributions.
>
> We respond to each of the concerns in detail below:
>
> * *No solution exists*: we will mention the assumption of the existence of a solution explicitly in the final version. Thank you for spotting this, but indeed, we are interested in optimization problems that have a solution.
> * Indeed, NGN requires non-negative objectives. However, we mention this explicitly, and therefore, we do not agree with the assessment that the comparison is unfair; for functions where NGN can be applied, it is more stable than SGD. Second, we could add an arbitrary constant to the objective without changing the problem; if the objective has a minimum (see above), then we can always apply NGN due to this. Third, the loss functions in ML, which are our main practical interests, are typically non-negative.
> * *Analysis is not optimal*: we would like to clarify that $\alpha_t$ can be arbitrary for all methods SGD, SPS, NGN and SPP. From the update rules, one can see that the **effective step size** of SPS and NGN can be different from $\alpha_t$, but our analysis accounts for that (otherwise, all methods would have the same stability index). As mentioned in late Section 1 (“Notation” paragraph), the quantity $\alpha_t$ is the user-specified step size. The main question is how sensitive the methods are wrt $\alpha_t$.
> * *No comparison to rates*: we discuss this in Appendix B.1. If the reviewer considers it necessary to highlight this more, we are happy to do so.
> * *On Figure 7*: this experiment shows that the range of (user-specified) learning rates (x-axis) where the performance is close to optimal matches the range where the theoretical bound is small. Therefore, we kindly disagree that the theory would not match the empirical result here. It is indeed true that the value of the bound might be different from the actual suboptimality, but this is mentioned (in bold) at the beginning of Section 5.
> * *Definition of stable*: the definition is through the behaviour of the stability index $\delta_t$, which is the main object of our analysis. We would point to section 4.5 for a more verbose explanation of this contribution.  In a nutshell, stability is defined through the sensitivity of the loss/bound with respect to (large) user-specified step sizes $\alpha_t$.
> * *Spike in bound for SGD*: this is due to the exploding gradient norm for large step sizes (see, for example, Figure 5). In case this point remains unclear, we are happy to provide further explanations.
> * *No practical applicability*: one practical implication of our paper is that it provides a clear theoretical explanation for why adaptive methods like SPS and NGN are favourable in terms of tuning cost (beyond the empirical evidence for this, which was established in prior works). We agree with the assessment that the current submission does not explore other, more novel, practical applications. However, many techniques are possible, for example, computing the stability index on-the-fly (which can be done essentially for free for SGD/SPS/NGN), and then adapting the step-size accordingly. However, as these techniques would require extensive benchmarking and combination with other standard techniques, we decided to leave it for future work and instead focus on the theoretical analysis for this paper.

---

> > ### Author Rebuttal · Reviewer_vs6x · 2026-04-03
> >
> > > Second, we could add an arbitrary constant to the objective without changing the problem; if the objective has a minimum (see above), then we can always apply NGN due to this.
> >
> > Finding such constants need the knowledge of $f_*$ which is same as solving the problem itself.
> >
> > > If the reviewer considers it necessary to highlight this more, we are happy to do so.
> >
> > I think they should be moved to main text.
> >
> > > Therefore, we kindly disagree that the theory would not match the empirical result here.
> >
> > I apologise for misunderstanding the plot. I agree with you. I will update my score accordingly.
> >
> > However, I think the practical applicability and novelty in proof techniques is still questionable.

---

> > > ### Author Response · Authors · 2026-04-03
> > >
> > > We are happy to hear that the concerns were fully resolved, and thank the reviewer for adjusting their score already.
> > >
> > > Regarding the comment on knowledge of $f^\*$, we kindly disagree: First, it would be sufficient to know a **lower bound** of $f^\*$, which is in general much less restrictive. Second, in machine learning, most loss functions of interest are lower bounded by zero (as they are coming from a negative log likelihood), which we can know a priori *without* knowing the solution to the problem.
> > >
> > > Finally, on this point, we would like to stress that the algorithm we study, especially SPS and NGN, are actually practical for machine/deep learning (see for example https://arxiv.org/abs/2508.15071, https://arxiv.org/abs/2305.07583). We think that this makes their theoretical study relevant, and also justifies the assumptions that we make.
> > >
> > > We would also like to point out that since the submission, we were able to extend our analysis to non-convex functions. See our response to reviewer TAu3 for a compact summary of these results (which will be included in the final version). We hope that this might convince the reviewer to reconsider the assessment regarding theoretical depth.

---

### Official Review · Reviewer_aBrE · 2026-03-04

**Soundness:** 2
**Presentation:** 4
**Significance:** 3
**Originality:** 3
**Overall Recommendation:** 5
**Confidence:** 4

**Summary:**

This work proposes a "stability index", based on the framework of model-based stochastic optimization. The dependence between the step size and the stability index relates to how robust the method is to the step size parameter. Analysis of this quantity is presented for SGD, SPS, NGN, and SPP methods. The paper demonstrates that the stability index of SPS, NGN, and SPP is smaller than that of SGD, indicating a better robustness to the step size parameter. Small-scale experiments support this claim.

**Compliance With Llm Reviewing Policy:**

Affirmed.

**Final Justification:**

I thank the authors and the reviewers.

At this point, my concerns were addressed, and I did not find the concerns raised by the other reviewers so far to be a reason for rejection. Hence, I keep my positive evaluation of this work.

**Key Questions For Authors:**

7. Perhaps consider discussing a natural problem class, like the square loss $f(x;z)=\frac12 ((x-x^\star)\cdot z)^2$, for which $2f(x;z)=\lVert \nabla f(x;z) \rVert^2$. If my back-of-the-envelope calculations are correct, the difference between SGD and SPS is very clear here.
8. Can AdaGrad-norm work with the modeling proposed in this work? I.e., when $\alpha_t=\alpha/\sqrt{\sum_{s=1}^t \lVert g_s \rVert^2}$? The main theorems still hold?
9. Line 226: In the context of SPS, the lower bound can be with respect to the infimum or with respect to the expected function $f$ (which is weaker). Is the assumption made in this work the standard for SPS, or are there guarantees using the weaker form?

**Limitations:**

yes

**Strengths And Weaknesses:**

**Strengths**

1. (Presentation) The paper is clearly written, the modeling framework and convergence guarantees are neat, and I enjoyed reading the paper.
2. (Significance) Studying the robustness of optimization methods is of value, and the presented framework and results provide a valuable tool and baseline for understanding this robustness.
3. (Originality) The stability index is a novel notion that yields the very nice-looking Theorems 3.3 and 3.4.

**Weaknesses**

4. (Soundness) A main claim of the paper is the improved stability index, which supports better robustness. The issue I see here is that if $\delta_t^{SPS} \leq \delta_t^{SGD}$, it does not mean that the dependence on the step size is better. E.g., if $\delta_t^{SPS} = 0.1 \cdot \delta_t^{SGD}$, it only indicates that perhaps SPS is better in general, not more robust. I think the authors could better support their claim by providing examples with basic class functions, e.g., the square loss $f(x;z)=\frac12 ((x-x^\star)\cdot z)^2$. I do believe the claims of the authors, I only indicate that they could be better supported.

5. (Significance, minor) The stability index is model dependent rather than algorithm dependent. This is fine for SGD, but other algorithms may not yield a supporting model. For example, the fixed $x_{t+1}=x_t$, and possibly adaptive algorithms.

6. (Significance, minor, no need to comment) Having experiments with modern scale would expose this work to a wider audience beyond the theory-oriented people in ICML. That said, the existing experiments are totally fine for a theory-focused work and I do not ask for any additional experiments in the rebuttal.

**Minor comments** (no need to respond)

- It would be appreciated if the authors could better indicate parts of the framework/results that were introduced by previous work. E.g., after introducing assumptions A1 and A2, indicating that they are used/introduced in Asi & Duchi (2019a). (From what I gather, everything "stability index" related is introduced by this current work.)
- Line 210: I suggest (though totally optional) including a concrete proof for Theorem 3.4 rather than saying it is identical to some other proof in another paper with modifications. The fact that modeling with $\delta_t$ abstracts the explicit analysis with $g_t$ is a nice touch.
- Line 221: There is no Theorem 9 in Schaipp et al. (2025, Thm. 9). After some work, there is one such Theorem 9 in the first version of their arXiv version (this is not the latest arXiv version of their work). In addition, please state which of your theorems translates to their result (I assume Theorem 3.4 from context, but please make it easier for the readers).



I overall enjoyed reading this work, currently supports it's acceptance, and would be happy if my main points will be sufficiently addressed.

---

> ### Author Rebuttal · Authors · 2026-03-30
>
> We would like to thank the reviewer for their detailed feedback and questions. In particular, we are very glad to hear that the reviewer enjoyed reading the paper and generally thinks that our contributions are interesting.
>
> In particular, we would like to thank the reviewer for the suggestions to improve the presentation of the paper, as well as for suggesting special cases to further clarify the theoretical results. We think that the changes based on these suggestions improved the quality of our submission already. Among these changes, we will also include new results that extend our framework to weakly convex functions (see response to Reviewer **TAu3** for more details).
>
> Answers to key questions:
>
> * *Relationship between $\delta_t$ comparison and robustness*: indeed, the main point is not to have an improvement of a constant factor, but a different **scaling** of $\delta_t$ with $\alpha_t \to \infty$. This indeed holds for SPS and NGN, as they do not scale linearly with $\alpha_t$, see for example Figure 1 right. We are happy to clarify further if needed.
> * *Special function classes*: Thank you for this suggestion! Indeed, for this special case, we can simplify the stability index for all methods. In particular,
> We obtain the following formulae, which further illustrate the advantage of SPS/NGN/SPP.
>
> \begin{align*}
>     \delta_t^{\texttt{SGD}} &= \frac{\alpha_t}{2}||g_t||^2, \\\\
>     \delta_t^{\texttt{SPS}} &\leq \min\\{\frac{\alpha_t}{2}, \frac{1}{2s_t^2}\\}||g_t||^2, \\\\
>     \delta_t^{\texttt{NGN}} &= \frac{\alpha_t}{2(1+\alpha_ts_t^2)}\||g_t||^2,  \\\\
>     \delta_t^{\texttt{SPP}} &= \frac{\alpha_t}{2}\frac{1}{1+\alpha_ts_t^2}||g_t||^2. \\\\
> \end{align*}
>
> We will include this in the final version.
> * *Extending to AdaGrad-norm*: This is an interesting suggestion. However, the model that results in AdaGrad norm would be $f_t + \frac{ \langle g_t, y-x \rangle}{\sqrt{\sum \|g_j\|^2}}$. For this model, it is not obvious how to satisfy assumptions (A1)-(A2). Another way to adapt the framework would be to allow matrix-induced norms in the proximal term of update (2) (similar to mirror descent). This is also how SPS can be extended to Adam/Adagrad in practice (see https://arxiv.org/pdf/2305.07583). However, obtaining such bounds becomes much harder.
> * *SPS lower bound*: The assumption we make is to know a lower bound of the stochastic loss $f(\cdot,s)$, as is usual for SPS analysis.

---

> > ### Author Rebuttal · Reviewer_aBrE · 2026-04-02
> >
> > I thank the authors for their response. My concerns were addressed, and I will raise my score.
> >
> > I would also like to respectfully point out that while Reviewer vs6x raises valid criticism on different parts of the work, in my humble opinion, most of these are minor and not a reason for rejection.
> >
> > The primary point for a major criticism that I think may be very reasonable is the novelty in theory compared to previous results for each algorithm. But in my opinion, the novelty is sufficient, even if it is on the weaker side.
> >
> > As for practical concerns, this is a theoretical paper that I think will be of interest to the theory community of ICML, and I think this is sufficient, even if the practical folks might find it totally impractical.

---

### Official Review · Reviewer_rdxB · 2026-03-08

**Soundness:** 3
**Presentation:** 3
**Significance:** 3
**Originality:** 3
**Overall Recommendation:** 4
**Confidence:** 4

**Summary:**

In this paper, the authors studies the impact of step size (learning rate) on the performance of several classes of algorithms. For this purpose, a quantity of stability index is proposed, and this index has been compared for several algorithms, including stochastic gradient descent (SGD), the stochastic Polyak step size (SPS), non-negative Gauss-Newtonsteps (NGN), and stochastic proximal point (SPP). The theory are developed for convex programs, and it is observed that similar behavior appear in non convex optimization as well.

**Compliance With Llm Reviewing Policy:**

Affirmed.

**Final Justification:**

The authors' response provide their underlying thinking on the issues I had with the paper, I thus maintain my moderate positive opinion on the paper.

**Key Questions For Authors:**

The stability index is a random variable obtained in the running of the algorithm, not some quantity that can be determined a priori, this seems to limit its usage. Have the authors considered any quantities that are more deterministic or can be decided with intensive engagement of the algorithms?

**Limitations:**

yes

**Strengths And Weaknesses:**

The paper is generally well-written, and step size is a critical problem in optimization. Theorem 3.3 and Theorem 3.4 are good observations.

The key quantity of stability index $\delta_t$ given in (3) is basically the difference of the objective values in (2) at $x_t$ and $x_{t+1}$ since $f(x_t,s_t)=f_{x_t}(x_t,s_t)$ by (A2). Thus, for convex function, it is not difficult to see that SPS,NGN and SPP will have smaller stability index since they are taking smaller steps which can be verified by checking the left column on page 2. Therefore, while it is a nice perspective to look at the stability index, I don't feel that results in the paper are revealing any deep insights.

---

> ### Author Rebuttal · Authors · 2026-03-30
>
> We would like to thank the reviewer for their feedback and their positive evaluation of our submission. Below, we respond to the key questions in more detail.
>
> * *No deep insights*: We agree that from the update rules of SGD/SPS/… one could already guess that some are more stable with respect to the choice of the step size. However, we would like to stress that one of our main contributions is to **prove this connection rigorously** (see Theorem 3.3/3.4 and the theoretical results of Section 4). Further, our contribution lies in identifying the quantity $\delta_t$ as the main factor for stability, which again, in our view, was not obvious at first sight.
> * *Practicality of stability index*: one way to use the stability index in practice is to compute it on-the-fly (which can be done essentially for free for SGD/SPS/NGN), and then adapt the step-size accordingly. For example, if the stability index grows too large, decrease the step size. The main focus of this submission is to derive the theoretical foundations and to show that the (a-posteriori) stability index is indeed reflecting the stable range of learning rates. We think that investigating practical strategies as the one described above are worth an independent paper, as the practical considerations usually pose additional challenges.

---

> > ### Author Rebuttal · Reviewer_rdxB · 2026-04-03
> >
> > Thanks for the authors for their response. I would like to maintain my score.

---

### Official Review · Reviewer_TAu3 · 2026-03-10

**Soundness:** 3
**Presentation:** 3
**Significance:** 2
**Originality:** 3
**Overall Recommendation:** 3
**Confidence:** 3

**Summary:**

The paper works to build theoretical understanding of why adaptive methods outperform SGD in stochastic optimization.
The core thesis of the paper is that several adaptive methods (stochastic Polyak, non-negative Guass newton, stochastic proximal point) are stable with respect to the scale of the step size, in contrast to SGD.
Doing so provides evidence on the robustness of these adaptive methods over SGD.
This gain in robustness is proven in convex examples, and is validated by experiments on training a ResNet on CIFAR 10 as well as convex examples that match the theory.

**Compliance With Llm Reviewing Policy:**

Affirmed.

**Final Justification:**

The authors propose a nice and novel framework for understanding robustness of adaptive methods vs SGD. Unfortunately I did not think the results of the paper are comprehensive enough for publication, but once fully fleshed out the paper could be a strong submission. The rebuttal did not address these concerns completely; their last response contains details only for the weakly convex case, and it is not a-priori clear how to generalize it to e.g. finding stationary points in the non-convex case.

**Key Questions For Authors:**

Is it possible to use the current proof strategy to derive any theory for non-convex examples, e.g. on the stability of adaptive methods vs SGD for finding a first order stationary point?

The experiments for training a ResNet on CIFAR 10 use a very large base learning rates $\alpha$ (like $\alpha \ge 1$ without warmup); this corresponds to step size 1 with constant learning rate schedule $\eta_t=1$, and thus step size $\eta=\alpha \cdot \eta_t \ge 1$ which is huge and much larger than what one would use in practice. This does not seem to be a comparison that is very relevant for practice, could the authors please clarify?

Are there adaptive methods that do not have a favorable stability index? It would be interesting to see what the theory says and how this matches experiments (even if the theory is not perfectly matching experiments, I think this would still be valuable to understand the limitations of the current theory, for future work to build off the theory in this paper, etc).

**Limitations:**

yes

**Strengths And Weaknesses:**

The paper aims to address an interesting and relatively original question regarding the robustness of adaptive methods vs SGD. The proofs seem correct, although I did not carefully check details in the appendix, and evaluation seems reasonable. The writing was ok, but I found the paper presentation a bit hard to follow at times. Specifically it was not obvious on first read how the $f_x$ in Sec 3 leads to the recursions that define SGD, SPS, NGN etc (as explained in Sec 4). It would have been much more motivated (and help the reader appreciate the contribution of the paper) to give some context in Sec 3 on how Lemma 3.2, Theorems 3.3 and 3.4 are a common framework that are applied to several algorithms (SGD and adaptive methods) in Sec 4. Also the proof of Theorem 3.4 basically follows from citing other results, however to be self contained it would be nice to at least rewrite the results being cited in the appendix.

As stated above, I believe the question is interesting way to justify and possibly select different optimization methods. The paper does a reasonable job of addressing this question, building theory in a convex setting, and validating the theory in experimental settings more relevant to ML. I also like how the paper is relatively short and clean, and more of the style of elegant insights rather than heavy computations. Considering the stability index leads to clean and simple proofs of the main results Lemma 3.2, Theorem 3.3, 3.4.

My biggest issue with the paper is that the contribution seems somewhat surface level, from the theory to the experiments. The theory only holds for convex examples, there are no bounds (even weaker results) that apply for non-convex examples. While in the analysis expressions on $\delta_t$ are derived for different algorithms, and those for adaptive methods are smaller than those for SGD, it is not clear which one is best in various different regimes or problems among the different $\delta_t$. The message seems more to be "these expressions are smaller than the corresponding expressions for SGD" (there is some study of the quantitative behavior of the expressions but I would like some more). Finally, the theory seems relatively simple and the proofs seem to follow from analogous rationale as typical convergence proofs for first order methods on convex losses. However, I think this is not a major weakness as the paper aims to address more conceptual questions.

The experiments similarly seem surface level. They only show a major gap between the loss and stability index for adaptive methods vs SGD for training a ResNet on CIFAR 10 for very large base learning rates $\alpha$ (like $\alpha \ge 1$ without warmup); this corresponds to step size 1 with constant learning rate schedule $\eta_t=1$ ($\eta = \alpha \cdot \eta_t$). This scale of step size seems much larger than what anyone would reasonably use in practice. It would also be good to dive deeper beyond just comparing to SGD: among the adaptive algorithms considered, what algorithms are more stable experimentally at various different learning rate scales, and how does this match the theory?

---

> ### Author Rebuttal · Authors · 2026-03-30
>
> We thank the reviewer for their comments and feedback. In particular, we are delighted to hear that the reviewer finds our contributions interesting and relevant. We hope that after addressing the concerns raised, the reviewer might consider a higher score.
>
> Below, we address each concern/question in detail.
>
> * *Presentation of the theory*: we highly appreciate your feedback, as this will allow us to improve upon the presentation. Indeed, the main methods are introduced on page 2, but not connected to the general model-based update (2). We will add one example for this connection earlier in Section 3 in order to make this clearer.
> * *Only convex theory*: first, let us say that we agree that this is restrictive, however many previous works have shown that phenomena from convex theory transfer well even to non-convex deep learning problems (for example, see https://arxiv.org/abs/2501.18965). Second, in the meantime, since submission, we have extended our analysis to weakly convex problems, which include the class of nonconvex, Lipschitz-smooth functions that are usually studied. The proof technique for this class is slightly more involved (based on Thm 4.3 in https://epubs.siam.org/doi/10.1137/18M1178244), which is why we had not included it in the first version. In short, the non-convex theory bounds the norm of the gradient of the Moreau envelope (a measure for stationarity), and has a similar structure to Thm 3.3 in the sense that the first term decreases with the sum of step sizes, and the second term is proportional to $\delta_t$. We are happy to discuss this in more detail and will of course, include these new results in the final version.
> * *This scale of step size seems much larger than what anyone would reasonably use in practice*: We kindly disagree with this. The scale of the optimal step size is optimizer-dependent and is usually much higher for SGD than for Adam or related methods. Further, our experiments (and many previous experiments, for example, in https://arxiv.org/abs/2305.07583, https://arxiv.org/pdf/2407.04358) prove that this learning rate is the one that results in the best validation loss, hence we do not see why the choice would be unusual.
> * *On adaptive algorithms*: Our framework can be extended to other adaptive methods that fit in the model-based framework (for example, for NGN this encompasses an entire family of methods, see Appendix D). Based on our results, this only requires the stability index. Extending the theory to adaptive methods such as Adagrad/Adam/… would be also nice to have, and therefore have clearly stated this in the limitations section. Regarding the empirical finding that SPS or NGN-type methods that are combined with Adam also lead to higher robustness than plain Adam, this has been extensively studied in previous work (see https://arxiv.org/abs/2305.07583 and arxiv.org/abs/2508.15071). Therefore, our conclusions about stability do transfer to adaptive methods, even though the theoretical proof is missing yet.

---

> > ### Author Rebuttal · Reviewer_TAu3 · 2026-04-01
> >
> > Thanks for the reply. My concerns r.e. the resulting step size have been resolved. My main issue with the paper however is unresolved: the paper needs more fleshing out before it is ready for publication. I think the idea is interesting and could be a nice submission down the line, but I believe the paper needs to be more comprehensive and in depth, as I stated in my original review. As such I'll keep my present evaluation.
> >
> > For example, having the theory extend (even straightforwardly) to the non-convex case would be a major improvement; I understand the reviewers claim it has been done, but I would like to see it in the manuscript itself. There are several more directions of inquiry that would add significant value: for instance, obtaining theoretical results for other adaptive algorithms like Adagrad, Adam, and comparing $\delta_t$ between different adaptive algorithms and seeing which one is best in different settings.

---

> > > ### Author Response · Authors · 2026-04-02
> > >
> > > We are glad to hear that the main remaining concern is depth of our contributions. As the reviewer mentioned that extending to non-convex functions would be a major improvement in that regard, below is the main result and base lemma for the weakly convex case. (As ICML does not allow to upload links to content other than tables and figures, we present a compact version here; the final paper will include all necessary details).
> > >
> > > **We hope that this can resolve the main remaining concern**, and therefore would kindly ask the reviewer again to reconsider the score.
> > >
> > > The proof technique is similar to the one in Section 4 of *Davis & Drusvyatskiy, Stochastic model-based minimization of weakly convex functions, SIAM Journal on Optimization, 2019*. However, we do not make the assumption of Lipschitz continuity (B4 in Davis & Drusvyatskiy 2019), and instead introduce the stability index $\delta_t$ in the proof. Therefore, this proof is not a trivial adaptation.
> > >
> > > We need to generalize assumptions A1 and A2 as follows:
> > >
> > > (B1) For any $x\in \mathbb{R}^d$ and $s\in \mathcal{S}$, the mapping $y\mapsto f_x(y,s)$ is $\lambda$-weakly convex for some $\lambda\geq 0$.
> > >
> > > (B2) We have $f_x(x,s) = f(x,s)$ for all $x\in \mathbb{R}^d$. Further, for all $s\in \mathcal{S}$ there exists $\rho_s \geq 0$ such that for all $x,y\in \mathbb{R}^d$ we have $f_x(y,s) \leq f(y,s) + \frac{\rho_s}{2}||x-y||^2$. Assume that $\rho := \mathbb{E}_s[\rho_s] < +\infty $.
> > >
> > > Note that (B1) and (B2) imply that $f$ is $(\rho+\lambda)$-weakly convex (Davis & Drusvyatskiy, Lemma 4.1)
> > >
> > >
> > > **Lemma** Assume that B1 and B2 hold. Let $\alpha_t \in (0, 1/\lambda)$. Let $x_t\in \mathbb{R}^d$ and denote by $\mathbb{E}_t$ the expectation with respect to the filtration generated by $x_1,\ldots,x_t$. Let $\bar{\rho} > \rho + \lambda$ and let $\hat{x}_t$ be the proximal operator of $(1/ \bar{\rho})f$ evaluated at $x_t$ (note: we only need $\hat{x}_t$ for analysis purposes, and never need to compute it). Then, we have
> > >
> > > $\mathbb{E}_t||x\_{t+1} - \hat{x}_t||^2 \leq ||x_t - \hat{x}_t||^2 - \frac{\alpha_t(\bar{\rho}-\rho-\lambda)}{1-\alpha_t\lambda}||x_t - \hat{x}_t||^2 + \frac{\alpha_t}{1-\alpha_t\lambda} \mathbb{E}_t[\delta_t].$
> > >
> > > *Proof*: Similar to the proof of Lemma 4.2 in D&D 19, but introducing the stability index instead of using Lipschitz continuity.
> > >
> > >
> > > We denote the Moreau envelope as $\mathrm{env}\_{f}^\alpha (x) := \min\_{y\in \mathbb{R}^d} f(y) + \frac{1}{2\alpha}||y-x||^2$.
> > >
> > > **Theorem** Assume that B1 and B2 hold, which in particular implies that $f$ is $(\rho+\lambda)$-weakly convex. Let $1\leq k \leq T \in \mathbb{N}$ and let the iterates $(x_t)\_{t\in \mathbb{N}}$ be generated by (2). Let $\alpha_t \in (0, 1/\bar{\rho})$ for all $t\in \mathbb{N}$, and for $\bar{\rho} > \rho + \lambda$. Denoting $\mathcal{E}_t:= \mathbb{E}[\mathrm{env}\_{f}^{1/\bar{\rho}}(x_t)]$, it holds
> > >
> > > $\sum\_{t=k}^{T} \frac{\alpha_t}{1-\alpha_t\lambda} \mathbb{E}||\nabla \mathrm{env}_f^{1/\bar{\rho}}(x_t)||^2 \leq \frac{2\bar{\rho}}{\bar{\rho}-\rho-\lambda} (\mathcal{E}_k - \mathcal{E}\_{T+1}) + \frac{\bar{\rho}^2}{\bar{\rho}-\rho-\lambda}\sum\_{t=k}^{T}\frac{\alpha_t}{1-\alpha_t \lambda} \mathbb{E}[\delta_t].$
> > >
> > > Setting $k=1$ we get a bound for the average gradient of the Moreau envelope, which serves as a measure of stationarity (see Davis & Drusvyatskiy, 2019). Compare also to Theorem 4.3 by Davis and Drusvyatskiy.
> > >
> > > *Proof*: Similar to the proof of Theorem 4.3 by D&D 2019 but using the above base lemma instead.
> > >
> > > Finally, **we want to highlight** that in the weakly convex case, the role of $\mathbb{E}[\delta_t]$ is exactly the same as in the convex case; it multiplies with $\alpha_t$, and gets divided but the sum of $\alpha_t$ (if we rearange the inequality of the theorem above). Hence, if $\mathbb{E}[\delta_t]$ scales sub-linearly with $\alpha_t$, the bound will be more stable!

---

### Decision · Program_Chairs · 2026-04-30

**Decision:**

Accept (regular)

**Comment:**

This paper studies an important theoretical question, namely, how to understand the robustness of stochastic optimization methods with respect to step size. The reviewers generally agreed that the submission is technically sound within its stated scope, and I do not view its focus on the convex setting as a fundamental limitation. Advancing understanding in this setting is already a meaningful contribution, especially for a theory-oriented audience. The paper also offers a clear comparison of several selected step-size rules using the proposed stability index, and this perspective was well received by multiple reviewers.

At the same time, I think the contribution should be interpreted with some caution. My reading is that the stability index may, in part, be a more abstract way of comparing one-step progress or one-step sensitivity across the selected methods, and it is not yet fully clear to me how much stronger the resulting conclusions are beyond that perspective. In particular, while the submission carefully compares selected step-size rules, it is less clear how far this analysis supports stronger comparisons of methods over their full optimization trajectory. Thus, although I do have some reservations about the ultimate scope of the new insight, I still find the question important, the analysis technically sound, and I thus recommend a *weak accept*.